# Elementwise Language Representation

## Abstract

We propose a new language representation method that generalizes all types of tokenization into a unified framework called elementwise language representation. This method represents each token using $\mathcal{N}$ low-dimensional byte embeddings, which are concatenated into a single vector. Using this framework, models can process text regardless of the tokenization applied. Most notably, by matching the number of attention heads in a Transformer architecture with $\mathcal{N}$, we can reduce its self-attention complexity proportional to the model size. This technique requires no architectural modifications of the backbone Transformer or additional overhead. Through experiment, we demonstrate that existing Transformer architectures trained within the proposed framework are improved in terms of efficiency, robustness and inference speed. These observations suggest the potential for an optimal pre-training objective built upon the elementwise language representation, guiding future works to focus on refining this approach.

## 1 Introduction

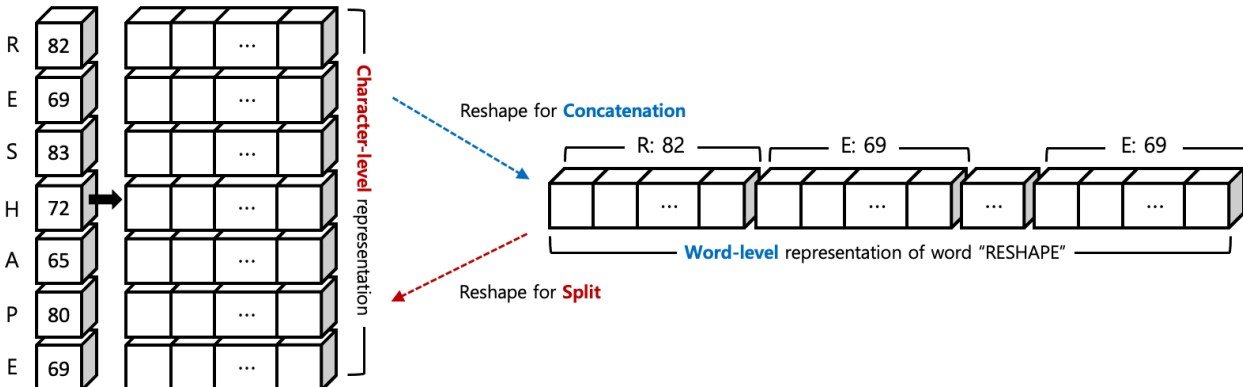

Figure 1: Visualization of the elementwise language representation. Any semantic unit is first encoded into a fixed number of bytes. The resulting bytes are projected into low-dimensional byte embeddings, and then concatenated back into a single vector, thereby forming the latent representation of the given semantic unit.

We understand text from various aspects of linguistics, but existing methodologies for language representation leverage tokenization which depends on a certain level of semantics exclusively, thereby ignoring many crucial parts of natural languages. Within these representations, text is encoded into a sequence of symbolic integers and transformed into latent embeddings with constant dimensionality (Bengio et al., 2000). This fundamental approach has been at the core of modern language representation, but cannot model the hierarchical structure between different levels of semantic units. This seemingly minor drawback leads to a major trade-off between vocabulary size and sequence length. Neural word-level models (Mikolov et al., 2013; Pennington et al., 2014) effectively maintain reasonable sequence length by aggregating several characters into a symbolic word token, but create a challenge in balancing out-of-memory and out-of-vocabulary problems. Although character-level models (Graves, 2013; Kim, 2014; Xue et al., 2022) avoid vocabulary-related issues, they embed each character into the same dimensional vector as word embeddings, thereby resulting in longer input sequences, increasing

computational complexity and long-range dependency. Subword-level tokenization (Sennrich et al., 2015; Wu et al., 2016; Kudo & Richardson, 2018) provides a good compromise between these two approaches, however, it still cannot model character-level information explicitly and faces the vocabulary problem in a multilingual setting again. While recent character-level models try to emulate word-level segmentation via downsampling input sequences (Tay et al., 2021; Clark et al., 2022; Godey et al., 2022), their acceptable downsampling rate is limited due to the issue of oversmoothing. We have now reached the time to explore a better representation.

In this paper, we propose a new framework called elementwise language representation, wherein each semantic unit is represented as a fixed number of low-dimensional byte embeddings. Each byte embedding corresponds to each letter of the semantic unit, which are concatenated into a single vector to form a latent representation.

Using this simple approach, we can represent any level of semantics of any type of natural language within a unified framework, regardless of the tokenization applied, utilizing only 256 low-dimensional byte embeddings. The character-level information of each semantic unit is modeled explicitly with its lexical properties. Because larger models with higher dimensionality can represent more letters jointly as a single latent embedding, the self-attention complexity on character sequences is reduced, in proportion to the number of byte embeddings per semantic unit. This implies that by simply matching the number of attention heads with the number of characters per each semantic unit, we can improve the computational efficiency of any Transformer (Vaswani et al., 2017) proportional to its model scale. No architectural modification or additional overhead is required.

This is the first part of the two-paper research on the elementwise language representation: we first introduce the framework for elementwise representation with theoretical and practical advantages in supervised learning in this article, and then explore an optimal pre-training objective across diverse scales in the follow-up study.

## 2    Background

In the real world, concepts are often composed of smaller sub-components hierarchically. Similarly, computers process data as sequences of bits, organized into bytes. The 118 elements serve as the fundamental building blocks of matter in this universe. Each element can be encoded using a single byte, which offers 256 possible values, enabling the representation of all known elements. Based on this intuition, we can represent *everything* within a unified framework called *elementwise representation* as follows:

1. Encode the given data into $\mathcal{N}$ bytes
2. Project each byte into a low-dimensional embedding
3. Concatenate $\mathcal{N}$ byte embeddings into a single latent vector

This latent representation separates the neural network architecture from the inductive bias of the data, and allows the data to be embedded while preserving its inherent hierarchical structure. See the examples below.

**Text modality.**    A semantic unit can be encoded into a sequence of UTF-8 bytes[1]. The encoded bytes are transformed into low-dimensional byte embeddings, and then concatenated into a single token representation.

**Image modality.**    An $(r, g, b)$ pixel is itself a length-3 byte tuple, each in range of $[0, 255]$. 3 color channels are projected into 3 low-dimensional byte embeddings, then concatenated into a single pixel representation.

**Sensor modality.**    An $(x, y, z)$ coordinate in a LiDAR point cloud is a length-3 float tuple. Each coordinate can be encoded into 4 or 8 bytes [2] with following projection, then concatenated into a single representation.

Because the same set of elements on different combinations can result in different materials, naively arranging byte embeddings might not be sufficient. We can derive unique representations from an identical set of bytes, by differently weighting each of the $\mathcal{N}$ bytes utilizing self-attention with $\mathcal{N}$ heads. As a single representation embeds $\mathcal{N}$ bytes jointly, each corresponding to a concept or element, a Transformer using $\mathcal{N}$ attention heads can align sequences $\mathcal{N}$ times longer, while maintaining its self-attention complexity fixed. This implies that the computational efficiency of the Transformer architecture improves as models scale, in proportion to the

---

[1]A text can be encoded into UTF-8 bytes via `list(bytes("text/to/encode", "utf-8"))` in `Python3`.

[2]A floating point number can be encoded into 4 or 8 bytes via `struct.pack("format", float)` in `Python3`.

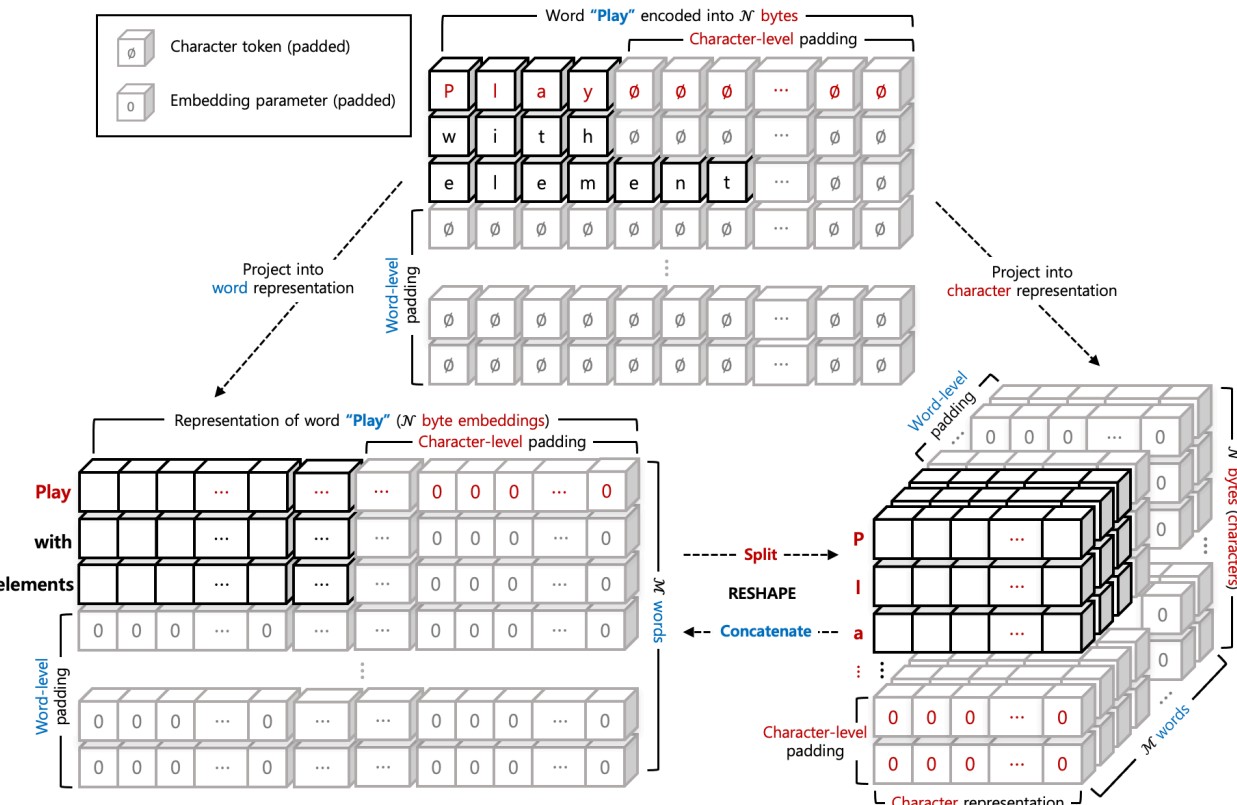

Figure 2: Visualization of the elementwise language representation framework. The proposed framework is designed to be generalized to all types of tokenization. Its explanation is most straightforward when applied with whitespace tokenization, wherein each semantic unit becomes a word consisting of $\mathcal{N}$ letters. First, the given text is tokenized into $\mathcal{M}$ words and each word is encoded into $\mathcal{N}$ bytes, with the hyperparameters $\mathcal{M}$ and $\mathcal{N}$. Words shorter than $\mathcal{N}$ are padded with integer zeros, longer ones are truncated. The byte encoding matrix of shape $(\mathcal{M}, \mathcal{N})$ is projected into a byte embedding tensor $(\mathcal{M}, \mathcal{N}, \mathcal{C})$, and then reshaped into a word embedding matrix of shape $(\mathcal{M}, \mathcal{D} = \mathcal{NC})$. $\mathcal{C}$ and $\mathcal{D}$ are the sizes of byte and word embeddings, respectively.

number attention heads. Using this simple approach, we believe that a *universal representation* for all kinds of modalities can be developed. By introducing the elementwise representation for the text modality in this paper, we take the first step towards this new research direction.

## 3  Elementwise Language Representation

The key idea behind the elementwise language representation is to represent each letter as a low-dimensional byte embedding and form a larger semantic unit by concatenating these byte embeddings into a single vector. Our approach is based on the assumption that the entropy of a semantic unit is proportional to the number of its letters. In other words, we use entropy as a criterion for determining the size of the byte embeddings.

Consider a word "*App_e*" with the missing spelling "*l*". The missing letter has low entropy since it is easily inferable. For a sentence with a missing word, such as "*_ brought a basket of apples,*" the entropy increases because the possible substitutions depend on the spellings of the subject. Similarly, the entropy will escalate when considering a missing sentence within a paragraph, as there are more cases to replace a sentence than a word within a sentence. Given this assumption, individual characters convey the simplest information and have the lowest entropy, enabling their representation using lower-dimensional embeddings than higher-level semantic units. See Figure 2 for the explanation below.

For an input sequence of $\mathcal{M}$ tokens (i.e., a sequence of semantic units), our method encodes each token into $\mathcal{N}$ bytes, with $\mathcal{N}$ as a hyperparameter. The selection of tokenization is irrelevant as our primary focus is on encoding each token into a byte sequence. Tokens with fewer than $\mathcal{N}$ letters are padded with integer zeros, while those with more are truncated. Shorter sequences are padded with blocks of $\mathcal{N}$ zeros. This technique facilitates the representation of any tokenization type within a unified framework, allowing neural networks to process input texts independently of the applied tokenization. Encoded $\mathcal{N}$ bytes are transformed into $\mathcal{N}$ low-dimensional byte embeddings and concatenated into a single vector through a tensor reshaping operation achievable in most contemporary deep learning libraries and frameworks. This operation is called *Reshaping*.

As $\mathcal{N}$ byte embeddings are concatenated to form a larger semantic unit, the resulting entropy increases again proportionally to $\mathcal{N}$. To reduce it back, we employ self-attention with $\mathcal{N}$ heads to attend on all $\mathcal{M}\mathcal{N}$ bytes, thereby concentrating the distributed probabilities on the most important semantic units. This approach is feasible since the representation and attention weights of each semantic unit are determined by both its $\mathcal{N}$ letters and the entire $\mathcal{M}\mathcal{N}$ letters via joint projection through a shared feed-forward layer. This operation is referred to as *Focus* wherein each of $\mathcal{N}$ attention heads explicitly focuses on latent space of byte embeddings rather than arbitrary subspaces of a symbolic embedding as in the original intuition of multi-head attention.

Although *Focus* attends on $\mathcal{M}\mathcal{N}$ byte tokens using a self-attention, its computational complexity is $O(\mathcal{M}^2)$, not $O((\mathcal{M}\mathcal{N})^2)$, fully ignoring the complexity of $\mathcal{N}$. This is because the representation of each semantic unit is composed of $\mathcal{N}$ byte embeddings, and each $n^{th}$ head of the $\mathcal{N}$ attention heads aligns each $n^{th}$ letter of all $\mathcal{M}$ semantic units with the following joint projection. Via backpropagation, we can assign attention weights to each $n^{th}$ letter in the corresponding $n^{th}$ attention head, jointly incorporating information from letters in different positions aligned in other attention heads. Regardless of the value of $\mathcal{N}$, which denotes the number of letters in each semantic unit, the complexity of *Focus* remains fixed to the number of semantic units, $\mathcal{M}$.

This leads to a reduction in self-attention complexity for an $\mathcal{M}\mathcal{N}$-length byte sequence, proportional to the choice of $\mathcal{N}$. As long as each byte embedding has sufficient dimensionality to represent individual characters, increasing $\mathcal{N}$ allows us to embed not only words but also higher-level semantic units such as phrases, sentences and even paragraphs into a single embedding. It indicates that larger models with higher-dimensional hidden layers can process longer texts than smaller ones with the same self-attention complexity. As any Transformer is itself stacked self-attention layers and the number of letters $\mathcal{N}$ also corresponds to the number of attention heads, larger Transformers with more attention heads can handle much longer sequences with reduced costs, compared to their smaller-sized equivalents without any architectural modification or additional overhead.

Within this simple "*Reshape*, then *Focus*" framework, our elementwise language representation embeds all kinds of text modalities jointly utilizing a tiny lookup table consisting of 256 low-dimensional byte embeddings and improves the computational efficiency of the Transformer architecture in proportion to the model scale. We believe this approach provides more natural and reasonable method for representing natural languages, in contrast to the traditional symbolic latent representations wherein different embedding tables are required for different types of tokenization and the computational costs of neural networks do not improve, no matter how large the models become.

## 3.1 Comparing the Representations

In this section, we elaborates the advantages of the proposed elementwise language representation compared to symbolic latent representations. We consider a $\text{BERT}_{\text{BASE}}$ (Devlin et al., 2018) as a backbone architecture.

**Versus character-level models.** Given an input text that consists of $\mathcal{K} = \mathcal{M}\mathcal{N}$ characters, it is converted into $\mathcal{K}$ character tokens and then transformed into a $(\mathcal{K}, 768)$ character embedding matrix via character-level model. The resulting self-attention complexity is $O(\mathcal{K}^2)$ and the character-level model adopts a $(256, 768)$ size embedding table, assuming a byte-level vocabulary for fair comparison with the elementwise representation.

By applying elementwise language representation, the $\mathcal{K}$ characters are encoded into $\mathcal{K}$ bytes. When $\mathcal{N} = 16$, the $\mathcal{K}$ bytes are projected into a $(\mathcal{K}, 48 = 768/\mathcal{N})$ byte embedding matrix, then reshaped into an embedding matrix of shape $(\mathcal{M} = \mathcal{K}/\mathcal{N}, 768)$ wherein each row represents a semantic unit consisting of 16 letters. This results in the $O(\mathcal{M}^2)$ self-attention complexity and the size of the elementwise embedding becomes $(256, 48)$. Hence, the model trained with elementwise language representation processes the same length byte sequence

with $\mathcal{N}^2$ times lower self-attention complexity, using $\mathcal{N}$ times fewer embedding parameters compared to the character-level model. Unlike recent studies that employ the sequence downsampling and rescale parameters to compensate for the downscaled embedding table (Tay et al., 2021; Clark et al., 2022), applying elementwise language representation does not entail modifying the backbone architecture or incurring additional overhead.

**Versus subword-level models.**    Given an input text comprising $\mathcal{M}$ words, a subword-level model converts it into $\mathcal{K} = \mathcal{M} + \alpha$ subword tokens and then projects them into a $(\mathcal{K}, 768)$ subword embedding matrix. The arbitrary length $\alpha$ is added to $\mathcal{M}$ due to the subword tokenization of each word. This results in the $O(\mathcal{K}^2)$ self-attention complexity and the subword-level model employs a subword embedding table of size $(30k, 768)$.

By applying elementwise language representation, the $\mathcal{M}$ words are converted into an $(\mathcal{M}, \mathcal{N})$ byte encoding matrix with each row representing a semantic unit of $\mathcal{N}$ bytes. As $\mathcal{N} = 16$, the matrix is transformed into a byte embedding tensor of shape $(\mathcal{M}, \mathcal{N}, 48 = 768/\mathcal{N})$, and then reshaped into a subword embedding matrix of shape $(\mathcal{M}, 768)$, where each row corresponds to a subword representation consisting of $\mathcal{N}$ byte embeddings. The resulting self-attention complexity is $O(\mathcal{M}^2)$ and the size of embedding table becomes $(256, 48)$. In this case, the reduction in computational complexity may vary depending on the counts of subword tokenization, denoted by $\alpha$. Since $\alpha$ tends to be larger for longer texts with more words, the improvement in computational cost will be significant proportionally to the sequence length $\mathcal{M}$. When $\mathcal{N}$ grows substantially beyond 16, a single token represents a larger semantic unit such as phrases, sentences or paragraphs with the same $O(\mathcal{M}^2)$ self-attention complexity, hence improving the computational efficiency of larger Transformers dramatically.

## 3.2    Implementation Details

The elementwise representation of the $m^{th}$ semantic unit that consists of $\mathcal{N}$ letters is implemented as follows:

$$e_m \leftarrow \text{Concat}(e_m^{(0)} \oplus g^{(0)}, ..., e_m^{(\mathcal{N}-1)} \oplus g^{(p)}) \oplus f^{(m)}$$

Consider that the input text is tokenized into $\mathcal{M}$ semantic units as $(S_0, ..., S_{\mathcal{M}-1})$. The $m^{th}$ semantic unit $S_m$ is encoded into $\mathcal{N}$ bytes as $S_m \leftarrow (B_0, ..., B_{\mathcal{N}-1})$. The number of the special tokens, such as [PAD] and [CLS], is added to each byte integer. If there are a total three special tokens, each byte is 3 greater than its original value. Shorter semantic units $S_m$ than $\mathcal{N}$ are padded with integer zeros, longer units are truncated.

The byte encoding of each $S_m$ is projected into a byte embedding sequence $(e_m^{(0)}, ..., e_m^{(\mathcal{N}-1)})$, where $e_m^{(n)} \in \mathbb{R}^{\mathcal{C}}$, and concatenated back into a single vector $e_m \in \mathbb{R}^{\mathcal{D}=\mathcal{N}\mathcal{C}}$ which constitutes the latent representation of $S_m$. The concatenation of the byte embeddings can be achieved via tensor reshaping from $(\mathcal{M}, \mathcal{N}, \mathcal{C})$ to $(\mathcal{M}, \mathcal{N}\mathcal{C})$.

Additionally, we can add two positional embeddings, $g^{(p)} \in \mathbb{R}^C$ and $f^{(m)} \in \mathbb{R}^D$, to $e_m^{(n)}$ and $e_m$ respectively, in order to manually encode positions that can be attended by the *Focus* operation. These embeddings are referred to as focus embeddings. The global focus embedding $g^{(p)}$ encodes the position of $n^{th}$ byte embedding in the $m^{th}$ semantic unit. The index $p$ is computed by the equation $p = (m \times \mathcal{N}) + n$, where $m \in [0, \mathcal{M})$ is the absolute position of each $m^{th}$ semantic unit, and $n \in [0, \mathcal{N})$ is the relative position of the $n^{th}$ letter in a semantic unit. On the other hand, the local focus embedding $f^{(m)}$ jointly encodes the positions of $\mathcal{N}$ letters in the $m^{th}$ semantic unit. Although the concept of elementwise language representation is well implemented without focus embeddings, we found that these are helpful for the stability and quality of model convergence.

Since the *Focus* on a sequence of $\mathcal{N}$-letter semantic units can be executed via Transformer with $\mathcal{N}$ attention heads, implementing the elementwise language representation entails simply replacing the existing embedding table of any Transformer model with 256 $\mathcal{C}$-dimensional byte embeddings. We abstract these byte embeddings with the following tensor reshaping operation from $(\mathcal{M}, \mathcal{N}, \mathcal{C})$ to $(\mathcal{M}, \mathcal{N}\mathcal{C})$ into an integrated embedding table called elementwise embedding. No other components of the backbone Transformer architecture are modified.

# 4    Experimental Setup

## 4.1    Dataset

We use patent documents published by the United States Patent and Trademark Office (USPTO) to evaluate different Transformers trained with the elementwise language representation in multi-label text classification.

Table 1: Reduced standard deviations of dataset splits used in the experiment. SD is the standard deviation between all CPC codes (the class labels), and $SD_{Major}$ refers to the standard deviation between CPC codes dominating over 90% of the entire labels in training examples. Standard deviations in all three dataset splits are significantly reduced after the proposed relabeling, thus implying that the class imbalances are alleviated.

| Labeling | Train | | | $2015_A$ | | | $2015_B$ | | |
|---|---|---|---|---|---|---|---|---|---|
| | Samples | SD | $SD_{Major}$ | Samples | SD | $SD_{Major}$ | Samples | SD | $SD_{Major}$ |
| Original | 1.9M | 21k | 35.4k | 14.5k | 1.8k | 3k | 15.4k | 1.9k | 3.2k |
| Relabeled | | 12.6k | 21k | | 1.1k | 1.9k | | 1.2k | 2k |

This custom dataset consists of 664 class labels, each representing a specific technical category of any patent. It features a highly imbalanced class distribution and includes a wide range of domain-specific lexicon related to each class label. A patent is a multimodal document written by expert applicant and maintained by patent office which is reliable. It consists of a title, abstract, claim, descriptions with figures and a set of classification symbols called Cooperative Patent Classification (CPC) codes. We utilize the claim section of each document as input text and use its associated CPC codes as labels. The objective of patent classification is multi-label prediction of the classes among 664 technical categories for the given patent claim that is usually a long text.

A single CPC code consists of five hierarchical levels of Section, Class, Subclass, Main Group and Subgroup. Among these, we use the slice from Section to Subclass as label. For instance, consider the CPC code A01N 53/12, which consists of [Section: A, Class: 01, Subclass: N, Main Group: 53/00, Subgroup: 53/12]. In this case, we utilize the concatenated slice [Section: A, Class: 01, Subclass: N], resulting in the label A01N. Each patent is assigned a sequence of CPC codes, with the primary CPC code indicating the main invention, and subsequent codes providing additional context. Hence, directly one-hot encoding the CPC codes may lead to skewed labels, as an identical CPC code can be used both as the primary and a subsequent code within the same label. For example, two different labels [A01B, G06Q, A01B], [G06Q, A01B] have an identical one-hot encoding [A01B:1 , G06Q: 1]. This wrong labeling confuses models to think of different patents as the same.

To address the order-dependent meaning of CPC codes and tackle the skewed label distribution, we prepend *First* to the primary CPC code and *Later* to the subsequent codes. The above labels [A01B, G06Q, A01B] and [G06Q, A01B] are reassigned as [*First*-A01B, *Later*-G06Q, *Later*-A01B] and [*First*-G06Q, *Later*-A01B], respectively. These labels result in two separate encodings [1, 0, 1, 1] and [0, 1, 1, 0], where the placeholder is [*First*-A01B, *First*-G06Q, *Later*-A01B, *Later*-G06Q]. Although this is in accordance with the official CPC documentation (USPTO, 2022), to the best of our knowledge, there have been no previous studies on patent classification that have followed this guidance. In addition to correcting the labels of the dataset for accurate evaluation, this modification mitigates dataset imbalance by dividing the frequency of each CPC code into *First* and *Later* categories, as evidenced by the reduced standard deviations between class labels in Table 1.

For training, we use patents published between 2006 and 2014 and ones published in 2015 for testing, divided into two splits $2015_A$ and $2015_B$. To prevent the models from overfitting on short claim segments that may not properly express the invention, we concatenate the first 20 claims of each patent into a single paragraph, rather than sampling the first ones only as in (Lee & Hsiang, 2020). All patent documents are utility patents published in the United States and collected using the BigQuery table called Google Patents Public Data[3].

## 4.2 Training Details

We trained models for text classification using the USPTO patent document dataset to benchmark robustness against domain specificity and imbalanced class distribution. Since the classification objective is multi-label, the binary cross-entropy loss with sigmoid activation is used for training. The threshold for prediction is 0.3.

As there is no known optimizer that is specifically tailored for patent data, we utilized the AdamW (Loshchilov & Hutter, 2017) which has demonstrated strong generalization for diverse applications and reduced sensitivity to different hyperparameter settings. It is configured using $\beta_1 = 0.9$, $\beta_2 = 0.999$, $eps = 1e - 8$ and $\lambda = 0.01$.

---

[3]https://github.com/google/patents-public-data

Table 2: Main differences between model configurations used in the experiments. Hyperparameters $\mathcal{M}$ and $\mathcal{N}$ refer to the number of tokens and of bytes comprising each token, respectively. $\mathcal{D}$ and $\mathcal{C}$ denote the sizes of token representations and their byte embeddings. Hyperparameter $\mathcal{A}$ means the number of attention heads.

| Model | Parameters Total | Embedding | $\mathcal{M}$ | $\mathcal{N}$ | $\mathcal{D}$ | $\mathcal{C}$ | $\mathcal{A}$ |
|---|---|---|---|---|---|---|---|
| BERT$_\text{EWE}$ | 87M (0.8x) | 12k (0.0005x) | 128 | 16 $(=\mathcal{A})$ | 768 $(=\mathcal{NC})$ | 48 | 16 $(=\mathcal{N})$ |
| BERT$_\text{ORIG}$ | 110M (1x) | 23M (1x) | 128 | 1 | 768 | 768 $(=\mathcal{D})$ | 12 |
| ALBERT$_\text{EWE}$ | 9M (0.7x) | 2k (0.0005x) | 128 | 16 $(=\mathcal{A})$ | 128 $(=\mathcal{NC})$ | 8 | 16 $(=\mathcal{N})$ |
| ALBERT$_\text{ORIG}$ | 12M (1x) | 4M (1x) | 128 | 1 | 128 | 768 $(=\mathcal{D})$ | 12 |
| CANINE$_\text{EWE}$ | 110M (0.8x) | 1M (0.05x) | 128 | 16 $(=\mathcal{A})$ | 768 $(=\mathcal{NC})$ | 48 | 16 $(=\mathcal{N})$ |
| CANINE$_\text{ORIG}$ | 130M (1x) | 25M (1x) | 128 | 1 | 768 | 768 $(=\mathcal{D})$ | 12 |

Table 3: Superiority of transformers trained with elementwise embedding in multilabel patent classification.

| Model | Speed(Samples/s) | $2015_\text{A}$ $F_1$ | Precision | Recall | $2015_\text{B}$ $F_1$ | Precision | Recall |
|---|---|---|---|---|---|---|---|
| BERT$_\text{EWE}$ | 629 (1.7x) | 64.30 | 66.02 | 62.66 | 63.94 | 66.55 | 61.53 |
| BERT$_\text{ORIG}$ | 366 (1x) | 63.68 | 65.59 | 61.82 | 63.35 | 67.16 | 59.95 |
| CANINE$_\text{EWE}$ | 339 (1.4x) | 64.30 | 65.86 | 62.82 | 63.95 | 66.43 | 61.64 |
| CANINE$_\text{ORIG}$ | 242 (1x) | 60.40 | 64.08 | 57.12 | 59.97 | 64.52 | 56.01 |
| ALBERT$_\text{EWE}$ | 638 (1.9x) | 63.18 | 65.84 | 60.73 | 62.91 | 66.47 | 59.71 |
| ALBERT$_\text{ORIG}$ | 338 (1x) | 63.15 | 65.82 | 60.70 | 62.79 | 66.36 | 59.59 |

We applied the learning rate decay to the initial learning rate of $2e-5$ without a warmup period. The batch size is set to 32. These small values are chosen to mitigate overfitting caused by the imbalanced training data. We trained all models for 10 epochs under a fully supervised learning setting without language pre-training.

### 4.3 Performance Measures

We leveraged three metrics: Precision, Recall and $F_1$ to evaluate the multi-label classification performance on the technical categories of patent documents. Precision captures how well a model identifies the unique label of each patent document. Recall measures how well a model distinguishes patents with different inventions.

The primary objective of patent classification is the optimization of the patent search engine, where Precision and Recall are equally important. Therefore, we use $F_1$ as the main metric to take into account both scores. We use these metrics to evaluate the model robustness to domain-specific and imbalanced patent documents.

Because our dataset exhibits extreme class imbalance with regard to the frequency of each technical category, all metrics are computed based on the micro-averaged scores over the entire class labels. The dataset contains 664 subclass-level CPC codes each corresponding to a single technical category and each code is divided into two labels by the prefixes *First* and *Later* (see Section 4.1), thereby creating 1,328 class labels to be averaged.

## 5 Results

In this experiment, we demonstrate the practical effectiveness of the elementwise language representation in terms of efficiency, robustness and inference speed. To isolate its effectiveness as a standalone representation technique from any enhancements introduced by various pre-training methodologies, we trained models from scratch, under fully supervised learning setting. Our goal is to show that existing Transformer architectures are improved by using the proposed representation, therefore the baselines are their original implementations.

To demonstrate the advantage in robustness by the elementwise language representation, we evaluate models on multi-label patent classification. Because the dataset used has a very long tail with regard to the frequency

of the class labels[4], the models should strive for high Recall to better classify the sparse classes without being biased towards the dominant categories in training examples, while maintaining a good balance with Precision to optimize the main metric $F_1$. This experimental setting is well-aligned with the previous studies on patent classification (Lee & Hsiang, 2020; Haghighian Roudsari et al., 2022), where established Transformers achieve poor recall with very high precision. The balanced scores in Table 3 are the result of mitigated class imbalance by the relabeling in Section 4.1, which further supports our usage of Recall with $F_1$ as a robustness measure.

We trained a $BERT_{BASE}$ replacing its embedding table with elementwise embedding (see Section 3.2), referred to as $BERT_{EWE}$ and compared with its original implementation $BERT_{ORIG}$ as a baseline. The only difference of $BERT_{EWE}$ from $BERT_{ORIG}$ is its use of elementwise embedding instead of the original subword embedding table and the number of attention heads, without any modification to the Transformer architecture (see Table 3). Via replacement from the $(30k, \mathcal{D} = 768)$ subword embedding table to the $(256, \mathcal{C} = 48)$ sized elementwise embedding, $BERT_{EWE}$ has 0.0005 times fewer embedding parameters than $BERT_{ORIG}$, resulting in 0.8 times smaller model size. As shown in Table 3, $BERT_{EWE}$ outperforms $BERT_{ORIG}$ in both Recall and $F_1$ in every test-set split. The improvement in $BERT_{EWE}$ is not due to the increased number of attention heads, because using 16 attention heads in $BERT_{ORIG}$ did not result in a meaningful difference in classification performance, only leading to unstable convergence. These enhancements are well-generalized to other Transformer models.

We generalize the elementwise language representation to two differently architected Transformers, ALBERT (Lan et al., 2019) and CANINE (Clark et al., 2022). As in $BERT_{EWE}$, their embedding tables are substituted with elementwise embedding table: the $(30k, \mathcal{D} = 128)$ subword embedding table of $ALBERT_{BASE}$ is replaced with the $(256, \mathcal{C} = 8)$ elementwise embedding and the $(16k, \mathcal{D} = 768)$[5] character embedding table of CANINE-C is swapped with the $(256, \mathcal{C} = 48)$ elementwise embedding table. Both outperform their baseline equivalents $ALBERT_{ORIG}$ and $CANINE_{ORIG}$ using 0.7 times and 0.8 times smaller model sizes respectively, maintaining their unique architectural components. This demonstrates the generalizability of elementwise representation. Though the improvement by the elementwise embedding is the largest in $CANINE_{EWE}$, it is still at the same level of classification performance as $BERT_{EWE}$ using 0.4 times fewer parameters (see Table 2 and 3). Hence, we need to consider whether the character encoding and sequence downsampling of the CANINE architecture are effective when the elementwise language representation is already applied to the Transformer architecture.

Since a single token representation consists of 16 byte embeddings via hyperparameter $\mathcal{N} = 16$, Transformers trained using elementwise language representation can process up to 16 times more letters than their original implementations at the same self-attention complexity. See Table 2 for reference. Compared to $CANINE_{ORIG}$ where 512 character embeddings are downsampled into $\mathcal{M} = 128$ input representations processing 512 letters with $O(\mathcal{M}^2)$ self-attention complexity, $CANINE_{EWE}$ processes $512 \times \mathcal{N} = 8192$ letters with the same $O(\mathcal{M}^2)$ complexity. In comparison to $BERT_{ORIG}$ and $ALBERT_{ORIG}$, which handle $4 \times \mathcal{M} = 512$ characters assuming 4 letters per subword, $BERT_{EWE}$ and $ALBERT_{EWE}$ both compute $\mathcal{M}\mathcal{N} = 2048$ letters with the same $O(M^2)$ self-attention complexity. In any case, as $\mathcal{N}$ grows, each token representation embeds more characters jointly, thus improving the computational efficiency of the backbone Transformer further, as explained in Section 3. So long as each byte embedding has sufficient dimensionality of $\mathcal{C}$ to encode the entropy of a byte, increasing $\mathcal{N}$ enables representation of larger semantic units with more letters, with lower computational burden. Based on the observation that the performance gain is less significant in $ALBERT_{EWE}$ utilizing the smallest $\mathcal{C} = 8$, we estimate the maximum number of characters a Transformer can embed into a single token representation to $\mathcal{N} = \mathcal{D}/8$, for embedding size $\mathcal{D}$. This is the reason why the elementwise language representation reduces the self-attention complexity of the Transformer proportional to its *scale* over the number of attention heads.

We applied the proposed language representation with whitespace tokenization in this experiment and suggest using it as the default approach also in practical applications, with a reasonable selection of the hyperparmeter $\mathcal{N}$. Even in a multilingual setting encompassing languages without whitespace-based word segmentation, we can simply increase $\mathcal{N}$ to make a single token embedding represent a larger semantic unit such as a sentence, or a paragraph. If a model cannot be sufficiently large to accommodate the necessary $\mathcal{N}$, we recommend to apply the tokenization-free approach as introduced in Section 5.1. Thanks to the removal of overheads from subword tokenization in $BERT_{EWE}$ and $ALBERT_{EWE}$ and from character-level encoding with downsampling in $CANINE_{EWE}$, every EWE model predict faster than its ORIG counterpart. Table 3 presents the improved

---

[4]Approx. 30% of dominant CPC codes account for 90% of the entire training examples.
[5]CANINE-C uses 8 $(16k, 96)$ sized Unicode character embedding tables.

Table 4: Effect of other tokenization on the convergence of transformers trained with elementwise embedding.

| Model | Tokenization Applied | $2015_A$ | | | $2015_B$ | | |
| --- | --- | --- | --- | --- | --- | --- | --- |
| | | $F_1$ | Precision | Recall | $F_1$ | Precision | Recall |
| BERT$_{EWE}$ | None | 60.01 | 63.77 | 56.67 | 59.80 | 64.49 | 55.75 |
| | Gradient | 64.14 | 65.91 | 62.45 | 63.75 | 66.41 | 61.29 |
| | Whitespace | 64.30 | 66.02 | 62.66 | 63.94 | 66.55 | 61.53 |

Table 5: Effect of focus embeddings on the convergence of transformers trained with elementwise embedding.

| Model | Ablated Components | $2015_A$ | | | $2015_B$ | | |
| --- | --- | --- | --- | --- | --- | --- | --- |
| | | $F_1$ | Precision | Recall | $F_1$ | Precision | Recall |
| BERT$_{EWE}$ | None | 64.30 | 66.02 | 62.66 | 63.94 | 66.55 | 61.53 |
| | Focus embeddings | 63.22 | 66.14 | 60.55 | 62.91 | 66.69 | 59.54 |

inference speed in EWE models when fully processing the test-set split $2015_A$ utilizing a batch size of 64 on a single RTX Titan GPU. Since there are no architectural differences between EWE and ORIG models and the tensor reshaping from $(\mathcal{M}, \mathcal{N}, \mathcal{C})$ to $(\mathcal{M}, \mathcal{D} = \mathcal{N}\mathcal{C})$ in the elementwise embedding entails trivial overhead compared to the subword tokenization and sequence downsampling, EWE models predict at least as quickly as ORIG models, if not slightly faster, using a batch size of 1. As a result, our proposed elementwise language representation enhances existing Transformer architectures with respect to robustness, efficiency and speed.

## 5.1 Effect of Tokenization

This section compares the efficacy of different tokenization applied with elementwise language representation.

We first train a BERT$_{EWE}$ using a tokenization-free approach. Unlike when applying whitespace tokenization, this version of elementwise language representation encodes text into an $\mathcal{M}\mathcal{N}$-length byte sequence directly. The resulting bytes are then projected into an $(\mathcal{M}\mathcal{N}, \mathcal{C})$ byte embedding matrix, then reshaped into an input embedding matrix of shape $(\mathcal{M}, \mathcal{D} = \mathcal{N}\mathcal{C})$. The hyperparameters used are the same as in the BERT$_{EWE}$ in Section 5 (see Table 2) and the two focus embeddings $g^{(p)}$ and $f^{(i)}$ are incorporated as introduced in Section 3.2. Due to the lack of whitespace segmentation, this tokenization-free BERT$_{EWE}$ performs much poorer in every metric for every test-set (see the first row of Table 4) even compared to the baseline model BERT$_{ORIG}$. Despite this performance degradation, it still achieves the same level of metrics as the CANINE$_{ORIG}$ having 0.5 times more parameters, thereby demonstrating again the effectiveness of the elementwise representation.

To provide an option for handling multilingual applications using small-sized backbones, we suggest applying gradient-based tokenization to the above tokenization-free BERT$_{EWE}$. For the hyperparameter $\mathcal{N}$, this model aggregates the $\mathcal{N}$-gram of each byte embedding via sum-pooling with softmax normalization. This approach aims to compensate for the missing segmentation by leveraging the information of neighborhood characters.

This gradient-based tokenization-free BERT$_{EWE}$ uses $\mathcal{N} = 8$, smaller than in the BERT$_{EWE}$ with whitespace tokenizer, to avoid oversmoothing of each byte embedding and does not leverage any type of focus embedding. Other hyperparamters did not change. This model recovers the same level of performance as the BERT$_{EWE}$ using whitespace tokenizer (see the second row of Table 4), proving that the performance enhancement from the elementwise language representation is not due to the increased number of attention heads, once again.

## 5.2 Effect of Focus Embeddings

As mentioned in Section 3.2, the main objective of focus embeddings is for stabilizing the model convergence. While elementwise language representation works well without these supplementary embeddings, its training can be somewhat tricky depending on the configuration of hyperparameters and random seeds. Incorporating the two levels of focus embeddings: global and local enables to ensure the reproducibility of model training.

In addition to providing enhanced stability, we found that focus embeddings also improve convergence quality. As shown in Table 5, $BERT_{EWE}$ shows better classification performance when trained with focus embeddings in terms of $F_1$ and Recall for both test-set splits. The slight decrease in Precision can be disregarded, because it does not substantially affect the main metric, $F_1$. Furthermore, the improvement in Recall is more valuable in patent classification, as the class imbalance of the dataset makes the model prone to being biased towards dominant class labels, so it is reasonable to see the enhancement in Precision alone as a result of overfitting.

We believe the efficacy of focus embeddings are not merely due to the naive addition of embedding parameters, as the increased amount is trivial[6], and adding more types of position embeddings was not found to be helpful.

## 6  Related Work

**Character-level Models**    The majority of past and current state-of-the-art and influential research in the field of natural language understanding has been built upon subword tokenization (Sennrich et al., 2015; Wu et al., 2016; Kudo & Richardson, 2018). This widespread adoption of subword tokenization can be attributed to the reasonable balance between robustness and efficiency, however, their inconsistency in tokenization and inefficient encoding of character-level information still remain as challenges to be addressed. Some follow-up studies have proposed improved technique for subword tokenization (Provilkov et al., 2019; He et al., 2020; Hiraoka et al., 2021; Wang et al., 2021) but many of these entail significant expenses and engineering efforts.

Character-level models have long been proposed as a promising alternative to the word-level representations. Although the chronic long-range dependency of character-level models (Sutskever et al., 2011; Graves, 2013; Zhang et al., 2015) have been solved by the adoption of the Transformer architecture as in (Belouadi & Eger, 2022; Xue et al., 2022), the quadratic complexity of self-attention mechanism has become a new bottleneck for character-level language representation. Some recently proposed studies attempt to address this problem by employing the sequence downsampling technique (Tay et al., 2021; Clark et al., 2022; Godey et al., 2022). These approaches are promising, but their downsampling rates are limited by the problem of oversmoothing. Additionally, the overheads and engineering challenges associated with their unique design choices can hinder their widespread adoption in various application areas. Our proposed elementwise language representation models any level of semantics maintaining the information of characters with lower self-attention complexity.

**Efficient Transformers**    The primary challenge of the Transformer architecture is to mitigate its quadratic self-attention complexity. Various promising approaches have been suggested. Liu et al. (2018) proposed to enhance self-attention complexity by computing attentions within chunked embedding matrices. Child et al. (2019) estimated full attention by mixing a sparse number of local attentions. Beltagy et al. (2020) extended this finding using a dilated sliding window technique to cover a wider range of attention. Zaheer et al. (2020) achieved similar results via attending on three different types of attentions: global, windowed, and random. Kitaev et al. (2020) alleviated memory complexity utilizing locally sensitive hashing and reversible residual layers Gomez et al. (2017). Wang et al. (2020) enhanced overall attention complexity to linear time reducing dimensionality of embeddings along the length axis. Katharopoulos et al. (2020) improved the self-attention complexity to linear time using kernel-based formulation and causal masking. To the best of our knowledge, the proposed elementwise representation is the first efficient method reducing the self-attention complexity of the Transformer proportional to the model size, without architectural modifications or additional overhead.

Approaches built upon architectural modifications other than attention mechanism have also been proposed. Lan et al. (2019) reduced the size of its BERT backbone significantly via parameter sharing between attention layers. The effectiveness of knowledge distillation (Hinton et al., 2015) has been demonstrated by (Jiao et al., 2019; Sanh et al., 2019; Tang et al., 2019). Approaches to improve inference-time efficiency include pruning away unimportant attention heads (Michel et al., 2019; Voita et al., 2019) or pruning blocks (Lagunas et al., 2021). As a standalone representation technique, our elementwise language representation can be potentially applied with those efficient modeling methodologies in conjunction, without harming their own contributions.

---

[6]Approx. 200k parameters are added for 128 local focus embeddings $f^{(i)}$ and 2048 global focus embeddings $g^{(p)}$.

## 7 Conclusion and Future Work

So far, we introduced a new language representation which embeds semantic units in an elementwise manner.

This representation offers the following advantages:

- It generalizes every tokenization into a unified framework
- It reduces self-attention complexity proportionally to the model scale
- It reduces the number of embedding parameters of a Transformer by under 1%

These are achieved without requiring architectural modification to the Transformer or additional overheads.

This representation remains the following challenges:

- It does not account for various linguistic components other than semantics
- It has not yet proposed an optimal pretraining objective for language modeling
- It has not yet been tested across various scales, applications, and learning paradigms

This representation suggests new research directions, which can be extended as follows:

1. Pretraining a language model that deals with all types of tokenization
2. Pretraining a language model with various levels of semantics simultaneously
3. Pretraining a multimodal language model understanding every modality using bytes

Recent groundbreaking studies (Brown et al., 2020; Raffel et al., 2020; Jaegle et al., 2021; Reed et al., 2022) have successfully demonstrated the potential of pretraining multimodal, multitasking models. By expanding their contributions with these research directions, computers may eventually be able to understand the world solely based on their native language, bytes. We explore these topics in follow-up studies in the near future.

## 8 Broader Implication

The framework for the elementwise language representation has the potential to be generalized for universal representation in multimodal settings. Consider a GPT-3 with 96 attention heads and $12,288$ dimensionality. Please note that these explanations on potential have not been validated via thorough experimental evidence.

When trained with elementwise representation, it processes $\mathcal{M}\mathcal{N} = 2,048 \times 96 = 196,608$ characters at once.

Since it uses embeddings with a dimensionality of $\mathcal{D} = 12,288$, a single byte embedding has a dimensionality of $\mathcal{C} = \mathcal{D}/\mathcal{N} = 128$. It can represent an $(r, g, b)$ pixel as a 384-dimensional embedding, with 128-dimensional byte embedding per color channel. As each token representation can embed 32 pixels jointly by $(\mathcal{D}/3\mathcal{C} = 32)$, it can process $65,536 = 32\mathcal{M}$ pixels via $\mathcal{M} = 2048$ tokens, which are three $(256 \times 256)$ RGB images at once.

Assuming 64-bit double precision floating point numbers, it can represent an $(x, y, z)$ coordinate in a LiDAR point cloud as an embedding with a dimensionality of $3072 = 3 \times 8\mathcal{C}$. Each coordinate $x$, $y$, and $z$ is encoded into 8 bytes according to the IEEE 754 standard, projected into an $(8, \mathcal{C} = 128)$ byte embedding matrix and then reshaped into a single latent representation of size $1024 = 8\mathcal{C}$. Representations of the three coordinates are concatenated into a single $3072 = 3 \times 1024$ sized embedding. In this way, it can process $8192 = 4 \times 2048$ coordinates in a LiDAR point cloud, where each token embedding represents 4 coordinate embeddings jointly.

Overall, the GPT-3 trained using the elementwise representation can process approx. $39k$ words, i.e., $196,608$ characters with 5 letters per word, three $(256 \times 256)$ RGB images and 8192 LiDAR coordinates without any architectural modification. Neither a CNN for image processing nor a specialized component will be required. A total three modalities are represented using a shared $(256, \mathcal{C} = 128)$ size byte embedding table. Since each of the 96 $\mathcal{C}$-dimensional attention heads exactly attends on the latent space of $\mathcal{C}$-dimensional byte embeddings, the self-attention complexity on the $\mathcal{M}\mathcal{N} = 196,608$ byte embedding sequence remains fixed to $O(\mathcal{M}\sqrt{\mathcal{M}})$.

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
