# OpenReview forum: "Elementwise Language Representation"
_TMLR — Rejected by TMLR_

### Review · Reviewer_i5mE · 2023-03-28

**Summary Of Contributions:**

The authors propose a new way of tokenizing that models the character-level information but does not increase the computational cost of doing self-attention. The construction comes from simple "reshape" and "focus" tricks, which make minimal modifications to the existing transformer architecture. The character-level embeddings that are learned are significantly smaller than the subword-level embeddings used for the standard architecture, and the proposed approach, albeit simple, seems promising.

The authors work on patent classification and also contribute a new way of defining classes for that tasks, which significantly improves the data imbalance problem of the dataset.



**Audience:**

Yes

**Broader Impact Concerns:**

No Broader Impact concerns.

**Claims And Evidence:**

Yes

**Requested Changes:**

Change A: The abstract is too technical and hard to follow. Maybe start with the motivation first and then go into the technical details. Without proper context, the reader will be lost.

Change B: Can you explain what is $u$ in $uv$ UTF-8 bytes, i.e. the first time you mention $uv$? Is $u$ the number of tokens?

Change C: Section 1 is incredibly hard to follow. It ends with a reference to Fig. 1, but I belive the reference should come much early in the text. There is a lot of notation and indexing, and a serious bookkeeping is required without having an access to a proper visualization. Please restructure Section 1 to make it more reader-friendly. Furthermore, you talk about $u$ and $v$ but this notation does not appear in Figure 1.

Figure 2 is really useful in addressing my questions. But it should come much earlier in the text.

Change D: Can you explain what changes you make in your model? Am I correct to understand that you only modify the embedding layer of the Transformer?

Change E: In Section 2, I sugged you omit the explanation of why you cover only 1 in this paper. I also advice against writing about a follow up study in this paper. Figure 2's caption also contains such a discussion, which I belive should be omitted, or referenced in conclusion as future work.

Change F: The "Proposition" in the text should be stated as intuition. Currently, the statement is too vague to be formalized as a mathematical Proposition with a Proof.

Change G: Can you comment on what happens when the the original models in Table 1 have the same number of heads as the EWE models, e.g. from 12 increase them to 16? Would the performance be better?

Minor A: When writing quotations in latex use `` and '' instead of " and ".

Minor B: no longer provide <- no longer provides.

Minor C: Waht does $p=(i\times j) + j$ stand for in your definition of $g^{(p)}$. I am confused about that index $p$. Could you explain?

Minor D: do not waste <- not to waste

Minor E: converge more stable <- converge more stably.

Minor F: possible to pertaining <- possible to pretrain.




**Strengths And Weaknesses:**

Strength A: The motivation and execution of the paper are quite natural.

Strength B: The transformer architecture does not have to change, only the embedding layer.

Weakness A: Comparison with baselines from the literature is missing. The method is evaluated only on patent classification and all the baselines are performed by the authors. This makes it difficult to judge the impact of the contribution because it is not compared with results from the literature, and also it is not tested on a wide range of tasks.

Weakness B: It seems to me that a hyperparameter sweep is missing for all the methods. I am not sure how the authors selected the hyperparameters, and for a fair comparison, I would like to see a more thorough investigation.

Weakness C: It is not clear whether the better performance comes from the increased number of heads. What would happen if all ORIG models have 16 heads per layer?

Weakness D: It is not clear that patent classification might be the killer app of your method. I would expect to see other tasks where the "material" embeddings would be helpful.

Weakness E: The abstract and introduction of the paper need to be rewritten to better guide the reader.

---

> ### Author Response · Authors · 2023-04-25
> **Response to reviewer i5mE [1/4]**
>
> We would like to thank you for your thorough review and careful advice to improve our paper.
> We are currently revising the manuscript based on your suggestions, and the following are clarifications in response to your comments.
>
> ### 1. Readability of the paper
> We acknowledge that the technical language used in the abstract and introduction may make it challenging for readers to comprehend our study. To enhance readability, we will make the following revisions in the updated manuscript:
>
> 1. We will provide intuitive motivation before explaining the technical details in both the abstract and introduction.
> 2. To improve the manuscript's structure, we will move Figure 2 to an earlier position.
> 3. We will provide clear definitions of u and v when first mentioning them.
> 4. We will correct all the typos you have pointed out.
>
> As you advised, we believe that improving the overall readability of the paper can be achieved by providing more intuitive explanations and by restructuring important sections. However, given that our proposed method is based on existing techniques with minimal modifications, it can be challenging to explain its distinctiveness while maintaining conciseness. Consequently, we may not have included some useful and detailed descriptions in the paper. To address this, we will first respond to your questions and supplement our clarifications with concrete examples wherever possible in the latter part of each response.
>
> ### 2. Investigation on the hyperparmeter selection
> Most of the training settings were based on the configurations of the $\mathrm{BERT_{BASE}}$ model provided in the original paper [(Devlin et al., 2019)](https://arxiv.org/abs/1810.04805). Although $\mathrm{ALBERT}$ [(Lan et al., 2019)](https://arxiv.org/abs/1909.11942) and $\mathrm{CANINE}$ [(Clark et al., 2022)](https://arxiv.org/abs/2103.06874) used the $\mathrm{LAMB}$ optimizer [(You et al., 2019)](https://arxiv.org/abs/1904.00962), we chose to use $\mathrm{AdamW}$ [(Loshchilov & Hutter., 2019)](https://arxiv.org/abs/1711.05101), that is known to be well-generalizable for various conditions for our downstream task of patent classification, as there is no strong evidence that $\mathrm{LAMB}$ is optimal for it. The hyperparameters for the $\mathrm{AdamW}$ are all the same as those used in the original $\mathrm{BERT_{BASE}}$ model. To alleviate the extreme class imbalance in the patent document dataset, we selected an initial learning rate of 2e-5, which aimed to prevent the models from overfitting too early or oscillating. Higher learning rates slowed down the convergence, while lower learning rates made the models underfit. We also used a small batch size of 32 to prevent models from being biased towards the majority classes. In summary, we collected a set of hyperparameters based on the configurations of the $\mathrm{BERT}$ paper, but conducted the actual tuning process equally for all models. If you require further clarification regarding our hyperparameter search, please do not hesitate to let us know.
>
> ### 3. Effect of the number of attention heads in $\mathrm{ORIG}$ models
>
> During the initial phase of our experiments, we had the same concern and tried to address it. We trained each model until half of the total epochs and stopped if it did not converge meaningfully, and all $\mathrm{ORIG}$ models with 16 attention heads were often stopped before the entire training is completed. While some models started to converge before reaching the threshold, they also did not achieve a performance that would enable a meaningful comparison with $\mathrm{EWE}$ models. We found this to be quite weird so that trained a $\mathrm{BERT_{BASE}}$ (i.e., $\mathrm{BERT_{ORIG}}$) with 16 attention heads until the final epoch without a convergence threshold, but it still failed to converge properly. Due to a busy research deadline, at that time, we have decided to prioritize the next step of experiments rather than investing more time in training $\mathrm{ORIG}$ models with 16 heads to achieve comparable performance to $\mathrm{EWE}$ models.
>
> Besides, the 16 attention heads of $\mathrm{EWE}$ models are set based on the fact that the majority of the English words have fewer than 16 letters. Both more and fewer attention heads than this resulted in a decrease in performance. However, this observation is valid only when training $\mathrm{EWE}$ models using whitespace tokenization. For instance, the $\mathrm{BERT_{EWE}}$ with 8 attention heads using a gradient-based tokenizer performs on par with a 16-headed one trained using whitespace tokenizer (please sea [Section 6.2.]). It even has fewer parameters than 16-headed $\mathrm{BERT_{EWE}}$ since it does not use focus embeddings, so we believe that the performance enhancements in $\mathrm{EWE}$ models do not come from the increased number of attention heads. If you still have any concerns regarding this aspect, please let us know.

---

> > ### Author Response · Authors · 2023-04-25
> > **Response to reviewer i5mE [2/4]**
> >
> > ### 4. Rigor of the mathematical Proof
> >
> > We agree that the statement is too vague to be formalized as a mathematical proposition with proof. We wanted to theoretically prove that the framework of elementwise representation “always” holds, but failed due to lack of mathematical background. The proposition in the current manuscript is a very loose description of what we intended to prove mathematically, so we will rewrite it as a theoretical intuition following your kind advice.
> >
> > Off topic, we would like to revisit this mathematical proof in future research. Our approach was to mathematically demonstrate that it is possible to approximate the ground-truth entropies of important letters (i.e., bytes) in concatenated token representations, by gradually updating their attention weights via stochastic gradient descent, using softmax in self-attention as an uncertainty measure of entropy, given sufficient amount of training data. If you have any guidance or insight on this matter, we would greatly appreciate it. Thank you.
> >
> > ### 5. Clarification of notations and implementation detail
> >
> > #### **Notation $u$ and $v$**
> > As you correctly understood, $u$ refers to the number of the entire “semantic units” in the given input text and $v$ indicates the number of letters composing each semantic unit. While there is no problem to understand $u$ as the number of tokens, we often preferred the term “semantic unit” in many parts of the paper where the meaning of a token can become ambiguous.
> >
> > #### **Notation $p$**
> > We apologize for the confusion caused by the typo in the formula of $p$, the superscripted index of global focus embedding $g^{(p)}$. The equation for calculating $p$ should be $p = (i\times v) + j$ when both $i$ and $j$ start from $0$, not $p = (i\times j) + j$. $i$ refers to the position of each token, $j$ denotes the position of each letter in a semantic unit. $v$ is the hyperparameter that sets the number of letters (i.e., byte embeddings) that make up a single token. Therefore, the range of $p$ is $[0, uv-1]$ and a total $uv$ global focus embeddings $g^{(p)}$s encode positions of $uv$ letters in the entire input text, where $u$ is the number of tokens (i.e., semantic units).
> >
> > For example, if there are 6 tokens each consisting of 3 letters, the global position of the first letter of the first token is calculated as $(0\times 3) + 0 = 0$, and $p$ of the last letter of the last token is computed as $(5\times 3) + 2 = 17$. Similarly, $p$ of the last letter of third token becomes $(2\times 3) + 2 = 8$. We will correct the typo and move the definition of $p$ to [Section 4.2. Implementation Details] clearly stating that it only holds on where $i$ and $j$ starts from $0$ in the revised manuscript. Thank you for your careful feedback that enabled us to make this correction.
> >
> > #### **Implementation of elementwise embedding**
> > Yes, you correctly understood the implementation of elementwise embedding. According to the proposed methodology, every token is represented as a horizontal concatenation of $v$ byte embeddings so that the embedding table for elementwise representation consists of only 256 low dimensional byte embeddings. The concatenation of byte embeddings is achieved via tensor reshaping (which is often cheaper in many deep learning frameworks than tensor concatenation). We abstract these two components into an  lookup layer called elementwise embedding. Since the following $Focus$ operation (weighting the entire $uv$ byte embeddings using $v$-headed self-attention) is performed at no cost by the parent transformer, applying elementwise representation is just to replace the embedding table of any transformer model with elementwise embedding. In practical implementation, we can accomplish this via one line of $\mathrm{Python}$ code as `model.embedding = elementwise_embed`.

---

> > > ### Author Response · Authors · 2023-04-25
> > > **Response to reviewer i5mE [3/4]**
> > >
> > > ### 6. Baselines from the literature and a wider range of downstream tasks
> > >
> > > We acknowledge that our proposed methodology could have been more extensively validated across a wider range of downstream tasks and directly compared to results from previous literature. Therefore, we are considering to provide a $\mathrm{GLUE}$ [(Wang et al., 2018)](https://arxiv.org/abs/1804.07461) benchmark of a much smaller size $\mathrm{BERT}$ (having approx. $\mathrm{4M}$ parameters) pre-trained using elementwise language representation. It shows better performance on 7 out of 9 $\mathrm{GLUE}$ scores than an equivalently sized $\mathrm{BERT}$ which is distilled by a larger teacher transformer [(Turc at al., 2019)](https://arxiv.org/abs/1908.08962). This is an initial result of our Part 2 study where we aim to develop an optimal language modeling for the proposed method. While it is not yet the best result achieved through an optimal pre-training objective, we would be happy to include it in the revised version of manuscript if it would be helpful for your further understanding of our research.
> > >
> > > Separately, we would like to clarify why we chose patent classification as the evaluation method in this study. While pre-training language models in an unsupervised manner and benchmarking its fine-tuning capacity on standard datasets as $\mathrm{GLUE}$, $\mathrm{SuperGLUE}$ [(Wang et al., 2019)](https://arxiv.org/abs/1905.00537), and $\mathrm{BigBench}$ [(Srivastava et al., 2022)](https://arxiv.org/abs/2206.04615) has become a de-facto standard in NLU/NLP research, such experiments require significant computing resources. As we had limited resources (only one $\mathrm{RTX}$ $\mathrm{Titan}$ instance with 24GB VRAM), we selected a fully supervised text classification task to demonstrate the validity of our proposed method. We found that using $\mathrm{USPTO}$ patent documents as a classification dataset can be helpful for benchmarking model robustness due to their vast domain-specific lexicon and imbalanced class distribution. It is important to note that we used patent documents solely to demonstrate the validity of elementwise language representation affordably, and not to make patent classification the primary application.
> > >
> > > While neural machine translation (NMT) is a reasonable way to evaluate models under supervised learning setting, it was not feasible for us given our computational limitations. Scaling down the NMT model and mini-batch to a trainable size would also be a possible approach but then it makes difficult to establish baselines from the literature again, as the original $\mathrm{Transformer}$ [(Vaswani et al., 2017)](https://arxiv.org/abs/1706.03762) was trained on eight $\mathrm{P100}$ GPUs with a total 128GB VRAM (16GB per instance), and most subsequent studies on Transformer-powered NMT have experimented with models of similar scale.
> > >
> > > Another probable approach is to apply efficient modeling and training strategies. However, many of these prioritize reproducing the performance of any pre-validated model more quickly and cost-effectively, rather than accelerating the development cycle of a new method. For example, $\mathrm{ELECTRA}$ [(Clark et al., 2020)](https://arxiv.org/abs/2003.10555) pre-trains a ¼ size $\mathrm{BERT_{BASE}}$ model on a single $\mathrm{V100}$ GPU in four days, but it still requires iterative experiments to generalize it to our proposed method. Unlike simply reproducing a pre-training with validated hyperparameters in four days, "iterating" a single development cycle that takes four days may not be considered a cost-effective experiment. Researchers often try to shorten this process by training the same model with different hyperparameter settings simultaneously on multiple GPUs, but this strategy was not affordable for us, too. Similarly, $\mathrm{Cramming}$ [(Geiping et al., 2022)](https://arxiv.org/abs/2212.14034) proposes a method for pre-training a full $\mathrm{BERT_{BASE}}$ on one 16GB GPU in one day, but it still cannot fully reproduce the original model's performance which makes it difficult for us to establish a robust baseline from existing literature, and also many experiments would be required to successfully apply it to our proposed representation method. Most of all, it was not publicly released at the time of our experiments.
> > >
> > > Apart from these facts, we have realized that it could confuse readers without enough background, so we will remove the mention of limited computational burden in the updated manuscript as you suggested. We appreciate your considerate advice once again.

---

> > > > ### Author Response · Authors · 2023-04-25
> > > > **Response to reviewer i5mE [4/4]**
> > > >
> > > > ### 7. Clarification of the main contribution
> > > >
> > > > While patent classification serves as a cost-effective means to demonstrate the validity of our study, it is not the ultimate design goal of elementwise representation. Our proposed method was designed to be a more efficient representation technique for general natural language processing and understanding, rather than being limited to specific downstream tasks.
> > > >
> > > > The main contributions of elementwise language representation are as follows:
> > > >
> > > > - It generalizes all types of tokenization into a unified framework
> > > > - It reduces self-attention complexity proportional to the number of attention heads
> > > >
> > > > Neither modification to the neural network architecture, nor additional overhead are required.
> > > >
> > > > Based on these advantages, we would like to explain the potential implication to LLMs’ energy efficiency of our research by applying the proposed elementwise language representation to $\mathrm{GPT}$-$3$ [(Brown et al., 2020)](https://arxiv.org/abs/2005.14165) as an example. While further research will be needed to prove the scalability of our method to such a massive scale, we believe this will help readers better understand our study in an intuitive manner.
> > > >
> > > > Consider a $\mathrm{GPT}$-$3$ $\mathrm{(175B)}$ trained with elementwise representation.
> > > >
> > > > Improvements in terms of computational cost are as follows:
> > > > - Each token consists of $96$ letters hence representing a paragraph, not a subword
> > > > - It has $96$ attention heads so processes $2,048\times 96 = 196,608$ tokens at once
> > > > - It processes $96$ times more tokens than the original $\mathrm{GPT}$-$3$
> > > > - It processes $6$ times more letters (assuming $16$ letters per words)
> > > >
> > > > at the same self-attention complexity of the original model.
> > > >
> > > > Enhancements in convergence speed and quality are as follows:
> > > > - It could be pre-trained at all levels of semantics (characters, words, paragraphs, etc.)
> > > > - It shares the same byte embedding table to learn all kinds of text modalities
> > > > - Multimodality and parameter sharing tends to improve convergence
> > > >
> > > > Although the exact vocabulary size of the original $\mathrm{GPT}$-$3$ has not been officially disclosed, let's assume that it has approximately $50,000$ English vocabularies, as in the case of $\mathrm{GPT}$-$2$ [(Radford et al., 2018)](https://arxiv.org/abs/2005.14165). Applying elementwise language representation, it will have a $(256, 128)$ byte embedding table for $128 = 12,288 / 96$ where $12,288$ is the hidden layer size and $96$ is the number of attention heads. The total number of embedding parameters is $32\mathrm{k} = 256\times 128$, which is 5e-5 times fewer than the original one’s $614\mathrm{M} = 12,288\times 50,000$. Actually, $\mathrm{GPT}$-$3$ uses massive multilingual vocabulary, so this reduction would be way more drastic in practical implementations.

---

### Review · Reviewer_u6Qv · 2023-04-08

**Summary Of Contributions:**

This paper proposes a variant on the Transformer each which instead of one embedding per token one uses one embedding per character and concatenating v consecutive characters per word to obtain a word embedding. v attention heads per Transformer position are used, with the i-th attention head selects the i-th character embedding of one of the words.

**Audience:**

No

**Broader Impact Concerns:**

I doubt that the proposed approach will worsen the existing ethial issues with Transformer-based LLMs.

**Claims And Evidence:**

No

**Requested Changes:**

See the Weaknesses section.

**Strengths And Weaknesses:**

Strengths:
- Given the success of Transformer architectures, it is worthwhile exploring a diversity of architectures that may bring computational or statistial advantages.
- Getting rid of the separately optimized decomposition of words into multi-character tokens would be interesting.

Weaknesses:
- It is not clear from the paper how to appropriately deal with the fact that different words have different lengths. This central fact explains previous approaches (using some form of per-word aggregation) that are quickly mentioned in the paper but NOT COMPARED experimentally. It is not even clear how this could work under the proposed approach, nor how the implementation deals with the issue.
- Experiments should compare on the same task & data with previously published results, not just of the regular per-token approaches but also the various charachter-based approaches.
- Experimental results should include an uncertainty estimate (p-values, confidence interval or standard deviation) to ascertain the statistical strength of the differences.
- The paper has pseudo-math. It is incorrect to call a Proposition and a Proof the ones on page 6. As a statement, the proposition is clearly not justified (for example, nothing in the proof tells us how entropy will be minimized). Even the statement of the proposition is not real math: it is vague (e.g. what does "important" mean, mathematically) and the so-called proof does not prove anything. Even the Assumption (top of page 6) does not make sense. Although we expect entropy to tend to increase with dimension (v), writing that it is actually linearly proportional is way too strong an assumption. Worse, the assumption is not really used in the proof in any formal way.
- The paper is full of vague and unclear statements. Even the introduction is unreadable (either too detailed for what is written to be understood before having read the rest of the paper, or not enough detailed for achieving sufficient clarity).
- Here are too many things that I did not understand. For example, why should we have p = i x j + j (bottom of p. 5). Many terms are used without being first defined (e.g., focus embeddings).
- It is not clear why the proposed approach even makes sense: why would the n-th character of each word be especially important? Why would the proposed approach be more "natural" than current approaches and why would it process data in a way that is more "reasonable" wrt sequence length (bottom of page 6)?
- Why would the proposed approach be expected to work better than previous approaches for character-level Transformers?

---

> ### Author Response · Authors · 2023-04-25
> **Response to reviewer u6Qv [1/5]**
>
> We would like to thank you for taking your time for reviewing our paper. We are currently editing the manuscript based on your feedback and the following are clarifications in response to your questions.
>
> ### 1. Inclusion of uncertainty estimates
>
> We would like to confirm that we have provided the reduced standard deviations between frequencies of class labels as an uncertainty measure in Table 2. $\mathrm{Total}$ refers to the std between all class labels, $\mathrm{Majors}$ denotes the std between class labels dominating over 90% of the entire labels in training examples. The mitigated imbalanced distribution was an incidental effect occurred during the relabeling process wherein correct the objective of patent classification and was not the main focus of our study, so we did not provide additional statistics as p-values and confidential interval. If you have any additional suggestions for improvement on this matter, please let us know.
>
> ### 2. Method for handling variable word lengths
>
> We apologize if any part of the explanation was unclear and caused confusion in understanding our proposed methodology. Nevertheless, we have provided explanations on how to handle the variability in word length in many parts of the current paper. We consistently mentioned throughout the paper that we represent each token by concatenating $v$ low-dimensional byte embeddings into a single vector. To supplement the technical details, in the caption of Figure 2, we clearly specified that tokens (e.g., words) shorter than $v$, consisting of fewer letters than $v$, are padded with integer zeros, and longer ones are truncated.
>
> In the case of training a $\mathrm{BERT_{BASE}}$ using elementwise representation with whitespace tokenizer and $v=16$ as in our experiments, each word is encoded into 16 bytes. Each byte is transformed into a 48-dimensional latent embedding for $48=768/v$, and $v$ byte embeddings are concatenated into a single 768-dimensional vector. Words shorter than $v=16$ are padded with integer zeros, otherwise truncated. Given an input sequence consisting of 128 words, it is encoded into a $(128, 16)$ bytes matrix wherein each row of shape $(1, 16)$ is an encoded word token, and then projected into a $(128, 16, 48)$ byte embedding tensor in which each sub-tensor $(1, 16, 48)$ is an embedded word token. This tensor is reshaped into a $(128, 768)$ input embedding matrix, which has the same shape as the input feature matrix of original $\mathrm{BERT_{BASE}}$.
>
> The implementation of this method is feasible in modern deep learning frameworks/libraries without requiring separate verification, and its effectiveness had been demonstrated through experiments on multi-label text classification practically.
>
> ### 3. Clarification of the theoretical background
>
> We understand your concern, but would like to clarify that we have provided sufficient context before defining our theoretical assumption.
>
> We use entropy as a criterion for embedding size selection. The proposed method represents any semantic unit (a word, phrase, sentence, etc.) by horizontally concatenating $v$ low-dimensional byte embeddings (each mapping to its spelling) into a single vector. Traditional character-level models use the same size latent embeddings as (sub)word-level models, but we believe that a much lower-dimensional embedding is sufficient for representing a character than a word embedding, since individual characters convey much simpler information than words do. It means that there are way fewer cases to encode at the character-level than at the higher-level of semantics so that the entropy of each letter (i.e., byte embedding) is the lowest. This is the main intuition of the assumption “The entropy of a semantic unit is proportional to the number of its letters.”
>
> The problem is that the entropy increases again as we build a larger semantic unit by concatenating $v$ byte embeddings. When there are $u$ token representations each consisting of $v$ byte embeddings as an input sequence, we can use self-attention with $v$ heads to lower the entropies back by concentrating attention weights onto more important bytes. Since each semantic unit consists of $v$ bytes and each of $v$ attention heads aligns each $n^{th}$ letter of all $u$ semantic units, we can process $uv$ byte tokens at the $O(u^2)$ self-attention complexity, fully ignoring the value of $v$. As self-attention complexity decreases proportional to the number of attention heads, larger transformers with more attention heads can process much longer sequence at the same computational complexity than smaller models (i.e., the acceptable input length become longer as models scale). This is the reason for why we believe that the proposed language representation is more “natural” and “reasonable” than traditional approaches, where the computational efficiency does not improve at all, no matter how large the model becomes.

---

> > ### Author Response · Authors · 2023-04-25
> > **Response to reviewer u6Qv [2/5]**
> >
> > ### 4. Rigor of the mathematical proof
> >
> > We acknowledge that the written statement is too vague to be formalized as a mathematical proposition with proof. We
> > wanted to theoretically prove that the framework of elementwise representation “always” holds, but failed due to lack of mathematical background. The proposition in the current manuscript is a very loose explanation of what we intended to prove mathematically, so we will rewrite it as a theoretical intuition instead.
> >
> > Off topic, we would like to revisit this mathematical proof in future research. Our approach was to mathematically demonstrate that it’s possible to approximate the ground-truth entropy of important letters (i.e., bytes) in concatenated token representations by gradually updating their attention weights via stochastic gradient descent, using softmax in self-attention as an uncertainty measure of entropy, given sufficient amount training data. If you have any guidance on this matter, we would greatly appreciate it. Thank you.
> >
> > ### 5. Clarification of notations
> >
> > We apologize for the confusion caused by the typo in the formula of $p$, the superscripted index of global focus embedding $g^{(p)}$. The equation for calculating $p$ should be $p = (i\times v) + j$ when both $i$ and $j$ start from $0$, not $p = (i\times j) + j$. $i$ refers to the position of each token, $j$ denotes the position of each letter in a token. $v$ is the hyperparameter that sets the number of letters (i.e., byte embeddings) that make up a single semantic unit. Therefore, the range of $p$ is $[0, uv-1]$ so that a total $uv$ global focus embeddings $g^{(p)}$s encode positions of $uv$ letters in the entire input text, where $u$ is the number of tokens (i.e., semantic units).
> >
> > For example, if there are 6 tokens each consisting of 3 letters, the global position of the first letter of the first token is calculated as $(0\times 3) + 0 = 0$, and $p$ of the last letter of the last token is computed as $(5\times 3) + 2 = 17$. Similarly, $p$ of the last letter of third token is $(2\times 3) + 2 = 8$. We will correct the typo and move the definition of $p$ to [Section 4.2. Implementation Details] clearly stating that it only holds on where $i$ and $j$ starts from $0$ in the revised manuscript. Thank you for your careful feedback that helped us to make this correction.
> >
> > We defined the notation for focus embeddings $g^{(p)}$ and $f^{(i)}$ in the early part of [Section 4. Methodology] and described its purpose in [Section 4.2. Implementation Details]. For the updated manuscript, we are considering explaining the definition of focus embedding earlier for better readability. We are grateful for your comment which will improve our writing. Thank you.

---

> > > ### Author Response · Authors · 2023-04-25
> > > **Response to reviewer u6Qv [3/5]**
> > >
> > > ### 6. Baselines from the literature and a wider range of downstream tasks
> > >
> > > We acknowledge that our proposed methodology could have been more extensively evaluated across a wider range of downstream tasks and directly compared to results from previous literature. Hence, we are considering to provide a $\mathrm{GLUE}$ [(Wang et al., 2018)](https://arxiv.org/abs/1804.07461) benchmark of a much smaller size $\mathrm{BERT}$ (having approx. $\mathrm{4M}$ parameters) pre-trained using elementwise language representation. It shows better performance on 7 out of 9 $\mathrm{GLUE}$ scores than an equivalently sized $\mathrm{BERT}$ which is distilled by a larger teacher transformer [(Turc at al., 2019)](https://arxiv.org/abs/1908.08962). This is an initial result of our Part 2 study where we aim to develop an optimal language modeling for the proposed method. While it is not yet the best result achieved through an optimal pre-training objective, we would be delighted to incorporate it in the revised version of manuscript if it would be helpful for your further understanding of our research.
> > >
> > > Additionally, we would like to clarify why we chose patent classification as the evaluation method in this study. While pre-training language models in an unsupervised manner and benchmarking its fine-tuning capacity on standard datasets as $\mathrm{GLUE}$, $\mathrm{SuperGLUE}$ [(Wang et al., 2019)](https://arxiv.org/abs/1905.00537), and $\mathrm{BigBench}$ [(Srivastava et al., 2022)](https://arxiv.org/abs/2206.04615) has become a de-facto standard in NLU/NLP research, such experiments require significant computing resources. As we had limited resources (only one $\mathrm{RTX}$ $\mathrm{Titan}$ instance with 24GB VRAM), we selected a fully supervised text classification task to demonstrate the validity of our proposed method. We found that using $\mathrm{USPTO}$ patent documents as a classification dataset can be helpful for benchmarking model robustness due to their vast domain-specific lexicon and imbalanced class distribution. It is important to note that we used patent documents solely to demonstrate the validity of elementwise language representation affordably, and not to make patent classification the primary application.
> > >
> > > While neural machine translation (NMT) is a reasonable way to evaluate models under supervised learning setting, it was not feasible for us given our computational limitations. Scaling down the NMT model and mini-batch to a trainable size would also be a possible approach but then it makes difficult to establish baselines from the literature again, as the original $\mathrm{Transformer}$ [(Vaswani et al., 2017)](https://arxiv.org/abs/1706.03762) was trained on eight $\mathrm{P100}$ GPUs with a total 128GB VRAM (16GB per instance), and most subsequent studies on Transformer-powered NMT have experimented with models of similar scale.
> > >
> > > Another promising approach is to apply efficient modeling and training methodologies. But many of these prioritize reproducing the training results of any pre-validated study more quickly and cost-effectively, rather than accelerating the development cycle of a new method. As an example, $\mathrm{ELECTRA}$ [(Clark et al., 2020)](https://arxiv.org/abs/2003.10555) pre-trains a ¼ size $\mathrm{BERT_{BASE}}$ model on a single $\mathrm{V100}$ GPU in four days, but it still entails iterative experiments to generalize it to our proposed method. Unlike simply reproducing a pre-training with pre-defined hyperparameters in four days, "iterating" a single development cycle that takes four days is not necessarily considered a cost-effective experiment. Researchers often try to shorten this process by training the same model with different hyperparameter settings simultaneously on multiple GPUs, but this strategy was not affordable for us, too. Similarly, $\mathrm{Cramming}$ [(Geiping et al., 2022)](https://arxiv.org/abs/2212.14034) proposes a method for pre-training a full $\mathrm{BERT_{BASE}}$ on one 16GB GPU in one day, but it still cannot fully reproduce the original model's performance which makes it difficult for us to establish a robust baseline from existing literature, and also many experiments would be needed to successfully apply it to our proposed representation method. Most of all, it was not publicly released at the time of our experiments.

---

> > > > ### Author Response · Authors · 2023-04-25
> > > > **Response to reviewer u6Qv [4/5]**
> > > >
> > > > ### 7. Clarification of the proposed method and its advantages
> > > >
> > > > The main contributions of the proposed elementwise language representation are as follows:
> > > >
> > > > - It generalizes all types of tokenization into a unified framework
> > > > - It reduces self-attention complexity proportional to the number of attention heads
> > > >
> > > > No modification to the parent neural network architecture and additional overhead are required.
> > > >
> > > > We would like to provide a step-by-step clarification of how the proposed framework can achieve such advantages:
> > > >
> > > > #### **Step 1. Reshape**
> > > > Given a sequence of $u$ tokens (i.e., semantic units), we encode each token into a sequence of $v$ bytes. The notable point is that it is not important at all, which tokenization was applied. Regardless of how the input text is tokenized, our only interest is to encode each token into $v$ bytes, where $v$ is a hyperparameter. By doing so, we can represent any type of tokenization within a unified framework, allowing neural networks to process input texts independently of the applied tokenization. Furthermore, this characteristic enables language models can be pre-trained at multiple levels of semantics from heterogeneous tokenizers “simultaneously” resulting in more robust and sophisticated representations of each semantic unit. We demonstrate this in our Part2 study.
> > > >
> > > > Encoded $v$ bytes are transformed into $v$ low-dimensional latent embeddings and then concatenated into a single vector. Since individual letters (i.e., bytes) convey much simpler information than any higher-level semantic unit such as a word, phrase, or sentence, it is natural to represent bytes as much lower-dimensional embeddings. It implies that there are fewer cases to encode, indicating the lowest entropy at the character-level. This is the core intuition behind why we use entropy as a criterion for selecting the size of the byte embeddings.
> > > >
> > > > #### **Step 2: Focus**
> > > > As $v$ byte embeddings are concatenated into a single vector to compose a larger semantic unit, the resulting entropy increases. To reduce this back, we use self-attention with $v$ heads to attend to all $uv$ bytes. This enables us to concentrate distributed probabilities on the most important letters (i.e., bytes), thus lowering the entropies of important semantic units. This is feasible because the representation and probability of each semantic unit are determined by both its $v$ spellings and the entire $uv$ letters, by being projected jointly via a shared feed-forward layer. It is worth noting that although the self-attention attends to $uv$ byte tokens, its computational complexity remains $O(u^2)$ regardless of the value of $v$. This is because the representation of each semantic unit is composed of $v$ byte embeddings, and each of the $v$ attention heads aligns each $n^{th}$ letter of all $u$ semantic units.
> > > >
> > > > This indicates that self-attention complexity decreases proportional to the number of attention heads $v$, thus enabling larger transformers to process longer sequences while maintaining their original computational complexities. For this reason, the acceptable sequence length increases as transformer models scale, and this is why we believe the proposed elementwise language representation is more “natural” and “reasonable” than the existing language representation strategies. Since the proposed method is a pure representation technique that is completely independent of the model architecture, the computational efficiency of the parent transformer can be further improved by upgrading its self-attention mechanism.
> > > >
> > > > Please feel free to ask any questions if there are still unclear parts.

---

> > > > > ### Author Response · Authors · 2023-04-25
> > > > > **Response to reviewer u6Qv [5/5]**
> > > > >
> > > > > ### 8. Clarification on comparison with existing methodologies
> > > > >
> > > > > We would like to clarify the advantages of our research in comparison to existing symbolic embedding methods for language representation.
> > > > >
> > > > > #### **Comparison with character-level representation**
> > > > > Consider an input text consisting of $x=uv$ letters. When using a $\mathrm{BERT_{BASE}}$ as the backbone architecture, character-level representations transform $x$ character tokens into a $(x, 768)$ embedding matrix with self-attention complexity of $O(x^2)$. In contrast, applying elementwise representation to $x$ letters transforms them into a $(x, 48)$ byte embedding matrix with $v=16$, as described in our paper, and then reshapes it to $(u, 768)$ for $u=x/v$ and $768=v\times 48$. This approach yields a self-attention complexity of $O(u^2)$ on $x=uv$ tokens, which ignores the value of $v$ and can process 16 times more tokens than character-level representations. Larger models with wider hidden layers and more attention heads, where $v$ grows further, can process even more tokens while maintaining their original self-attention complexity. It is worth noting that this enhancement in computational complexity is independent of both the implementation of self-attention. Therefore, the computational efficiency of an elementwise transformer can be further improved by the advances in self-attention mechanism and other architectural components.
> > > > >
> > > > > Recent studies, such as $\mathrm{CANINE}$ [(Clark et al., 2022)](https://arxiv.org/abs/2103.06874) and $\mathrm{Charformer}$ [(Tay et al., 2021)](https://arxiv.org/abs/2106.12672) employ a downsampling technique to shorten the input character sequence, which incurs additional computational and memory overhead. In contrast, our proposed method uses a single tensor reshaping operation, which results in negligible overhead and can be further reduced with technical tricks such as JIT compilation and asynchronous dispatch. Furthermore, while character- and byte-level models, including the aforementioned studies, rescale parameters for compensating the reduced embedding parameters, our transformers trained with elementwise language representation outperform their original implementations (i.e., baselines) without the need for parameter rescaling, as demonstrated in our experimental results. This advantage is well-maintained in the language modeling experiment, which will be incorporated into the updated manuscript.
> > > > >
> > > > > Elementwise representation differs from existing character-level representations in its use of non-symbolic embeddings that can incorporate both character and word-level information simultaneously, as well as any higher level of semantics for larger $v$. This characteristic allows for the learning of competitive, and even superior, representations without the need for additional operational enrichments or parameter complements. This theoretical advantage is practically demonstrated in our experimental results on multi-label text classification.
> > > > >
> > > > > #### **Comparison with subword-level representation**
> > > > >
> > > > > Given an input sequence of $u$ words, a subword-level $\mathrm{BERT_{BASE}}$ converts it into a $(x=u+a, 768)$ input embedding matrix, where $a$ is added to $u$ due to subword tokenization of each word. This results in a self-attention complexity of $O(x^2)$. By utilizing elementwise representation, we can represent the input sequence as a $(u, 16, 48)$ byte embedding tensor, where $48=768/v$ and $v=16$; its each $(1, 16, 48)$ sub-tensor represent an embedded word token. This is reshaped into a $(u, 768)$ embedding matrix with a self-attention complexity of $O(u^2)$ ignoring the complexity of $v$. Larger models with more attention heads can represent each token with more letters to construct a semantic unit, such as a phrase, sentence, or paragraph. Thus, while a large subword model still processes $x=u + a$ “subwords” at the $O(x^2)$ cost, a same size transformer trained using elementwise language representation handles $u$ larger semantic units such as “sentences” at the $O(u^2)$ self-attention complexity, for example.
> > > > >
> > > > > In contrast to symbolic embeddings of subword models, which cannot perfectly capture the information of individual characters, elementwise representation reflects both character and word-level information, as well as any other complex semantic units for larger $v$. We believe this feature improves the model robustness to domain specificity and noisy data. The theoretical advantage of this feature is practically demonstrated in our experimental results on multi-label text classification.
> > > > >
> > > > > If you feel that the distinctiveness of the proposed methodology is still unclear, we would greatly appreciate it if you could kindly point out where you think the issue lies. Thank you for your consideration.

---

### Review · Reviewer_RySd · 2023-04-16

**Summary Of Contributions:**

the submission proposed to represent words as a concatenation of vector embeddings of characters, and by setting the dimension of each attention head to be the same as that of the character embeddings, the attention module in the transformer now considers the association of a pair of characters at the same location of two words. The submission further claims that the modification is necessary for character-sensitive tasks, for example, patent classification, which is used in the submission to demonstrate the effectiveness of the proposed embedding method.

**Audience:**

Yes

**Claims And Evidence:**

No

**Requested Changes:**

Please address the weaknesses.

**Strengths And Weaknesses:**

*Strengths*

1. the proposed method is easy to implement.


*Weaknesses*

1. the method itself is simple enough that  explanations through "materials" are not necessary, and it could've been described in a much more succinct phrasing with existing diagrams.

2. the assumption that "we understand words by focusing on a few characters" comes without references, and I am not sure if it is even true. However, we do tend to understand sentences by focusing on a few words, and it has been studied in neurolinguists. If the authors believe that the original assumption is true, please add references.

3. "Assumption. For a v letters semantic unit x, H(x) ∝ v" , where H is the entropy. I am also not sure that this is true.

Without any condition on the context, the average number of characters per English word is roughly 4.5, and the distribution of the number of words in terms of the number of characters in each word has a very long tail. It means that there is an extremely high uncertainty when the missing word has 4 or 5 characters, and it is also very certain to us that, the word with a single letter can only be "i" or "a".With context, the uncertainty shrinks, but it wouldn't follow a simple monotonic trend.

4. the acceptable sequence length still depends on the number of words in a document, which is the same as other encoding methods. Therefore, I don't see the advantages of using this proposed approach.

5. the submission argues that due to the limit of their computational resources, they couldn't do pre-training on BERT. However, given the GPU device described in the paper, pre-training a standard BERT model on BookCorpus and Wiki103 --- these were used for pre-training BERT --- would still be doable within a reasonable amount of time. Since pre-trained models are prevalent and finetuning becomes cheaper every day, it would be useful to understand the impact of the proposed embedding method in pre-training.

6. The final point is regarding the performance. Table 3 doesn't seem to show that the improvement is significant since often the improvement is at the order of $10^{-3}$. The submission also didn't compare to standard NLP tasks, for example, GLUE or SuperGLUE.

---

> ### Author Response · Authors · 2023-04-25
> **Response to reviewer RySd [1/5]**
>
> Thank you for your detailed review and valuable comments on our paper. We are currently revising the manuscript based on your feedback, and following are clarifications in response to your questions.
>
> ### 1. Readability of the paper
>
> We agree with your advice that the readability of the paper can be improved in terms of terminology and descriptive aspects. We will reflect your feedbacks in the updated manuscript. Thank you for your thoughtful suggestion.
>
> ### 2. Clarification of the experimental results
>
> In the first of Section 6, Table 3 presents the superior classification performances of three differently architected transformers $\mathrm{BERT}$ [(Devlin et al., 2018)](https://arxiv.org/abs/1810.04805), $\mathrm{CANINE}$ [(Clark et al., 2022)](https://arxiv.org/abs/2103.06874), $\mathrm{ALBERT}$ [(Lan et al., 2019)](https://arxiv.org/abs/1909.11942) trained with the proposed elementwise language representation ($\mathrm{EWE}$ models) over their original implementations, the baseline models ($\mathrm{ORIG}$ models). The improved $\mathrm{F_{1}}$ on multi-label patent classification is $0.62$ in $\mathrm{BERT_{EWE}}$, $3.9$ in $\mathrm{CANINE_{EWE}}$, $0.03$ in $\mathrm{ALBERT_{EWE}}$. The only case showing an enhancement at the order of 10e-3 is $\mathrm{ALBERT_{EWE}}$. All $\mathrm{EWE}$ models outperformed their $\mathrm{ORIG}$ baselines despite using 20% (approx. $\mathrm{20M}$) fewer parameters, so even if the performance improvement is small or equivalent, it can still be considered a remarkable accomplishment.
>
> It is particularly noteworthy that our performance enhancements were achieved through a simple reshaping of input data, without the need for additional overheads, engineering efforts, or modifications to the backbone transformer architectures. The only difference between our $\mathrm{EWE}$ models and the $\mathrm{ORIG}$ models is the use of 256 low-dimensional byte embeddings in the $\mathrm{EWE}$ models. These embeddings are abstracted into an embedding lookup layer called "elementwise embedding", which uses under 1% of the embedding parameters of the original models. We ensured that all our $\mathrm{EWE}$ models and $\mathrm{ORIG}$ models shared the same architecture and configurations in the main text. Hyperparameter tuning was also performed consistently for all models. Therefore, we believe that our experimental comparison is fair, and the improvements in our results are sufficiently robust to demonstrate the validity of our research. However, if you have any remaining concerns about this matter, please do not hesitate to let us know.

---

> > ### Author Response · Authors · 2023-04-25
> > **Response to reviewer RySd [2/5]**
> >
> > ### 3. Clarification of the theoretical background
> >
> > #### **Clarification of the theoretical assumption**
> > We understand your concern, however, we would like to clarify that our theoretical background is not based on the assumption that the entropy of a sampled English word is proportional to the number of its letters. Rather, we utilize entropy as a criterion for embedding size selection and the concept of semantic units are not limited to words in our assumption.
> >
> > In the proposed framework of elementwise language representation, we encode a single semantic unit into a sequence of $v$ bytes, which are subsequently transformed into low-dimensional byte embeddings. These embeddings are then horizontally concatenated into a single vector, which serves as the latent representation of the semantic unit. Since individual characters contain much simpler information than words do, thus indicating that the entropy is the lowest at the character-level, we can represent letters (i.e., bytes) using way lower-dimensional embeddings than higher-level semantic units such as words, phrases and sentences. When $v$ letters (byte embeddings) compose a larger semantic unit by being concatenated into a single latent vector, its entropy grows again which means that the number of cases to be encoded increases. This is consistent with the fact that there are many more ways to substitute one sentence for another within a paragraph than to replace one word with another within a sentence. The core idea is that the larger the number of characters that make up the semantic unit, the more diverse the possible substitutions can be when expressing a similar meaning, which can be interpreted as high entropy. This interpretation is distinct from using entropy as a measure of uncertainty for the probability of sampling English words of a certain length, as you mentioned in your feedback.
> >
> > #### **Clarification of the intuition on the $Focus$ operation**
> >
> > As we concatenate $v$ byte embeddings to form larger semantic units, the resulting entropy increases. To address this issue, we can use a self-attention mechanism with $v$ heads to attend to all $uv$ byte embeddings, which allows us to concentrate probabilities on the most important semantic units; where $u$ is the number of tokens in the input text and $v$ is the number of letters (i.e., bytes) in each token. This is possible because the representation of each token is determined by both its own $v$ byte embeddings and the entire $uv$ byte embeddings. By backpropagating through the network, we can maximize the attention weight of the  important semantic units, thus minimizing their entropies.
> >
> > Just as a kind of intuition, this can be linked to our ability to capture the meaning of a word by focusing on only the most morphologically salient letter. We stated this in our paper twice as “It is similar to that we often catch the meanings of words by focusing only some morphologically noticeable spellings, so we call this operation focus” in [Section 1. Introduction; page. 2] and “It is quite similar to that we often read text inferring the meanings of words by focusing on a few important letters” in [Section 4.1 Elementwise Embedding; page. 6]. We appreciate your feedback, but would like to clarify that the sentence "We understand words by focusing on a few characters" is not present in our paper and does not accurately convey our intended meaning. Our intuition highlights our "ability" to estimate the meaning of a word by focusing on certain spellings that are morphologically more important in determining its interpretation. This is distinct from proposing a specific neuro-linguistic mechanism involved in human word understanding. One possible reference from cognitive psychology on this intuition is “Raeding Wrods With Jubmled Lettres: There Is a Cost.” (Rayner et al., 2006) wherein authors have analyzed that the letters in the middle of a word have smaller effects on determining its meaning. While our method may not perfectly align with their findings at the point that our method “learns” which letters should be more weighted via gradient descent, it seems that referring this may be helpful for readers when enough context is provided. We are considering a way to incorporate this citation with clear background into our manuscript concisely. Thank you for your thoughtful comment .

---

> > > ### Author Response · Authors · 2023-04-25
> > > **Response to reviewer RySd [3/5]**
> > >
> > > ### 4. Baselines from the literature and a wider range of downstream tasks
> > >
> > > We acknowledge that our proposed methodology could have been more extensively evaluated across a wider range of downstream tasks and directly compared to results from previous literature. Hence, we are considering to provide a $\mathrm{GLUE}$ [(Wang et al., 2018)](https://arxiv.org/abs/1804.07461) benchmark of a much smaller size $\mathrm{BERT}$ (having approx. $\mathrm{4M}$ parameters) pre-trained using elementwise language representation. It shows better performance on 7 out of 9 $\mathrm{GLUE}$ scores than an equivalently sized $\mathrm{BERT}$ which is distilled by a larger teacher transformer [(Turc at al., 2019)](https://arxiv.org/abs/1908.08962). This is an initial result of our Part 2 study where we aim to develop an optimal language modeling for the proposed method. While it is not yet the best result achieved through an optimal pre-training objective, we would be delighted to incorporate it in the revised version of manuscript if it would be helpful for your further understanding of our research.
> > >
> > > Besides, we would like to clarify why we did not utilize language modeling to demonstrate our proposed methodology. While pre-training language models in an unsupervised manner and benchmarking its fine-tuning capacity on standard datasets as $\mathrm{GLUE}$, $\mathrm{SuperGLUE}$ [(Wang et al., 2019)](https://arxiv.org/abs/1905.00537), and $\mathrm{BigBench}$ [(Srivastava et al., 2022)](https://arxiv.org/abs/2206.04615) has become a de-facto standard in NLU/NLP research, such experiments require significant computing resources. As our available resources were limited (only one $\mathrm{RTX}$ $\mathrm{Titan}$ instance with 24GB VRAM), we instead selected a fully supervised text classification task to demonstrate the validity of our study. We found that using $\mathrm{USPTO}$ patent documents as a classification dataset can be helpful for benchmarking model robustness due to their vast domain-specific lexicon and imbalanced class distribution. It is important to note that we used patent documents solely to demonstrate the effectiveness of elementwise language representation affordably, and not to make patent classification the primary application.
> > >
> > > As you commented, one promising way for language modeling with low computational resources is to apply efficient modeling and training methodologies. However, many of these prioritize reproducing the training results of any pre-validated model more quickly and cost-effectively, rather than accelerating the experimental cycle for developing a new method. As an example, $\mathrm{ELECTRA}$ [(Clark et al., 2020)](https://arxiv.org/abs/2003.10555) pre-trains a ¼ size $\mathrm{BERT_{BASE}}$ model on a single $\mathrm{V100}$ GPU in four days, but it still entails iterative experiments to generalize it to our proposed method. Unlike simply reproducing a pre-training with pre-defined hyperparameters in four days, "iterating" a single development cycle that takes four days is not necessarily considered a cost-effective experiment. Researchers often try to shorten this process by training the same model with different hyperparameter configurations simultaneously on multiple GPUs, but this strategy was not affordable for us, too. While $\mathrm{Cramming}$ [(Geiping et al., 2022)](https://arxiv.org/abs/2212.14034) proposes a method for pre-training a full $\mathrm{BERT_{BASE}}$ on one 16GB GPU in one day, it still cannot fully reproduce the original model's performance which makes it hard for us to set a robust baseline from previous literature, and also many experiments would be necessary to successfully apply it to our proposed representation method. Furthermore, it was not publicly released at the time of our experiments. Although we could also consider pre-training a tiny-sized transformer, we were not sure if there are any studies that can be used as strong baselines for purely pre-training minimal transformers without relying on knowledge distillation. To the best of our knowledge, other approaches on training efficiency also share similar context. If you happen to know a better way, we would greatly appreciate it if you could share it with us.

---

> > > > ### Author Response · Authors · 2023-04-25
> > > > **Response to reviewer RySd [4/5]**
> > > >
> > > > ### 5. Clarification on comparison with existing methodologies
> > > >
> > > > We would like to explicate the advantages of our research in comparison to the existing language representation methods.
> > > >
> > > > #### **Comparison with character-level representation**
> > > > Consider an input text consisting of $x=uv$ letters. When using a $\mathrm{BERT_{BASE}}$ as the backbone architecture, character-level representations transform $x$ character tokens into a $(x, 768)$ embedding matrix with self-attention complexity of $O(x^2)$. In contrast, applying elementwise representation to $x$ letters transforms them into a $(x, 48)$ byte embedding matrix with $v=16$, as described in our paper, and then reshapes it to $(u, 768)$ for $u=x/v$ and $768=v\times 48$. This approach yields a self-attention complexity of $O(u^2)$ on $x=uv$ tokens, which ignores the value of $v$ and can process 16 times more tokens than character-level representations. Larger models with wider hidden layers and more attention heads, where $v$ grows further, can process even more tokens while maintaining their original self-attention complexity. It is worth noting that this enhancement in computational complexity is independent of the implementation of self-attention. Therefore, the computational efficiency of an elementwise transformer can be further improved by the advances in self-attention mechanism and other architectural components.
> > > >
> > > > Recent studies, such as $\mathrm{CANINE}$ [(Clark et al., 2022)](https://arxiv.org/abs/2103.06874) and $\mathrm{Charformer}$ [(Tay et al., 2021)](https://arxiv.org/abs/2106.12672) employ a downsampling technique to shorten the input character sequence, which incurs additional computational and memory overhead. In contrast, our proposed method uses a single tensor reshaping operation, which results in negligible overhead and can be further reduced with technical tricks such as JIT compilation and asynchronous dispatch. Furthermore, while character- and byte-level models, including the aforementioned studies, rescale parameters for compensating the reduced embedding parameters, our transformers trained with elementwise language representation outperform their original implementations (i.e., baselines) without the need for parameter rescaling, as demonstrated in our experimental results. This advantage is well-maintained in the language modeling experiment, which will be incorporated into the updated manuscript.
> > > >
> > > > Elementwise representation differs from existing character-level representations in its use of non-symbolic embeddings that can incorporate both character and word-level information simultaneously, as well as any higher level of semantics for larger $v$. This characteristic allows for the learning of competitive, and even superior, representations without the need for additional operational enrichments or parameter complements. This theoretical advantage is practically demonstrated in our experimental results on multi-label text classification.
> > > >
> > > > #### **Comparison with subword-level representation**
> > > >
> > > > Given an input sequence of $u$ words, a subword-level $\mathrm{BERT_{BASE}}$ converts it into a $(x=u+a, 768)$ input embedding matrix, where $a$ is added to $u$ due to subword tokenization of each word. This results in a self-attention complexity of $O(x^2)$. By utilizing elementwise representation, we can represent the input sequence as a $(u, 16, 48)$ byte embedding tensor, where $48=768/v$ and $v=16$; its each $(1, 16, 48)$ sub-tensor represent an embedded word token. This is reshaped into a $(u, 768)$ embedding matrix with a self-attention complexity of $O(u^2)$ ignoring the complexity of $v$. Larger models with more attention heads can represent each token with more letters to construct a semantic unit, such as a phrase, sentence, or paragraph. Thus, while a large subword model still processes $x=u + a$ “subwords” at the $O(x^2)$ cost, a same size transformer trained using elementwise language representation handles $u$ larger semantic units such as “sentences” at the $O(u^2)$ self-attention complexity, for example.
> > > >
> > > > In contrast to symbolic embeddings of subword models, which cannot perfectly capture the information of individual characters, elementwise representation reflects both character and word-level information, as well as any other complex semantic units for larger $v$. We believe this feature improves the model robustness to domain specificity and noisy data. The theoretical advantage of this feature is practically demonstrated in our experimental results on multi-label text classification.
> > > >
> > > > If you remain unclear about the advantages of our proposed method, we would be happy to provide additional explanations or examples to clarify any points.

---

> > > > > ### Author Response · Authors · 2023-04-25
> > > > > **Response to reviewer RySd [5/5]**
> > > > >
> > > > > ### 6. Clarification of the proposed method and its advantages
> > > > >
> > > > > The key contributions of the proposed elementwise language representation can be summarized as follows:
> > > > >
> > > > > - It generalizes all types of tokenization into a unified framework
> > > > > - It reduces self-attention complexity proportional to the number of attention heads
> > > > >
> > > > > The parent neural network architecture does not need to be modified, and no additional overhead is required.
> > > > >
> > > > > We would like to offer a detailed explanation of how the proposed framework can attain these advantages through a step-by-step approach.
> > > > >
> > > > > #### **Step 1. Reshape**
> > > > > Given a sequence of $u$ tokens (i.e., semantic units), we encode each token into a sequence of $v$ bytes. The notable point is that it is not important at all, which tokenization was applied. Regardless of how the input text is tokenized, our only interest is to encode each token into $v$ bytes, where $v$ is a hyperparameter. By doing so, we can represent any type of tokenization within a unified framework, allowing neural networks to process input texts independently of the applied tokenization. Furthermore, this characteristic enables language models can be pre-trained at multiple levels of semantics from heterogeneous tokenizers “simultaneously” resulting in more robust and sophisticated representations of each semantic unit. We demonstrate this in our Part2 study.
> > > > >
> > > > > Encoded $v$ bytes are transformed into $v$ low-dimensional latent embeddings and then concatenated into a single vector. Since individual letters (i.e., bytes) convey much simpler information than any higher-level semantic unit such as a word, phrase, or sentence, it is natural to represent bytes as much lower-dimensional embeddings. It implies that there are fewer cases to encode, indicating the lowest entropy at the character-level. This is the core intuition behind why we use entropy as a criterion for selecting the size of the byte embeddings.
> > > > >
> > > > > #### **Step 2: Focus**
> > > > > As $v$ byte embeddings are concatenated into a single vector to compose a larger semantic unit, the resulting entropy increases. To reduce this back, we use self-attention with $v$ heads to attend to all $uv$ bytes. This enables us to concentrate distributed probabilities on the most important letters (i.e., bytes), thus lowering the entropies of important semantic units. This is feasible because the representation and probability of each semantic unit are determined by both its $v$ spellings and the entire $uv$ letters, by being projected jointly via a shared feed-forward layer. It is worth noting that although the self-attention attends to $uv$ byte tokens, its computational complexity remains $O(u^2)$ regardless of the value of $v$. This is because the representation of each semantic unit is composed of $v$ byte embeddings, and each of the $v$ attention heads aligns each $n^{th}$ letter of all $u$ semantic units.
> > > > >
> > > > > This indicates that self-attention complexity decreases proportional to the number of attention heads $v$, thus enabling larger transformers to process longer sequences while maintaining their original computational complexities. For this reason, the acceptable sequence length increases as transformer models scale, and this is why we believe the proposed elementwise language representation is more “natural” and “reasonable” than the existing language representation strategies. Since the proposed method is a pure representation technique that is completely independent of the model architecture, the computational efficiency of the parent transformer can be further improved by upgrading its self-attention mechanism.
> > > > >
> > > > > Please let us know if there are still unclear parts.

---

### Author Response · Authors · 2023-05-09
**Dear Editor and Reviewers,**

Thank you for your patience. We have submitted the final revised manuscript. We received significant feedback regarding the readability of our paper, and acknowledge the need for improvements. As a result, we spent considerable time revising our manuscript. While the content and contributions of the paper remain the same, we have rewritten every part of the manuscript.

As almost every section has been modified, we have not specifically marked the changes in the manuscript. We have consolidated important content into each section, making it possible to grasp the key points within one page. Unnecessary sections have been removed or merged, important figures have been updated, and their positions have been moved to the front.

We have eliminated confusing intuitions and formulas in our methodology description and rewritten it to focus on the most reliable arguments. We have thoroughly reviewed the experimental setup and provided a comprehensive analysis of the results. The only unresolved issue is the part about language modeling. As mentioned in a previous response, we considered providing some of the language modeling results from subsequent research, but that would require revealing the core methodology of that work. Since a significant part of follow-up research is already underway, we have decided not to provide the language modeling content in this paper, as originally planned.

Instead of adding new experimental results, we focused on clearly conveying the existing content and providing thorough practical evidence for our claims. Additionally, we have included some of the background and ideas that were planned to be included in the follow-up paper in the "Background" and "Broader Implications" sections. We have made the notation of standard deviations more explicit and added a chapter comparing the differences between our methodology and existing methods.

Thanks to the opportunity you provided for us to have our paper reviewed and your valuable time spent on the review, we were able to significantly improve the quality of our manuscript. Once again, we sincerely appreciate your help.

---

### Decision · Action_Editors · 2023-07-06

**Recommendation:** Reject

**Comment:**

The authors improved the paper after the initial review but did not address major concerns w.r.t. the provided evidence in the form of experimental evaluation (The authors removed some limited/incorrect theoretical claims in the revised version).

The authors initially wanted to add additional experiments, but then did not, however, this seems a critical limitation. I would encourage the authors to consider combing this work with their follow-up work, to create a strong single submission either to this or another venue.


**Audience:**

The general direction and idea of new and more efficient word embeddings in the context of language models such as transformers would be of interest to a large audience, however, the paper does not demonstrate that the proposed idea is effective as discussed above, limiting the audience to people who want to verify if this idea is actually working or not.

**Claims And Evidence:**

The authors claim e.g. in the abstract that "Through experiment, we demonstrate that existing Transformer architectures trained within the proposed framework are improved in terms of efficiency, robustness and inference speed.". However, as also discussed by all reviewers the experimental evidence is limited to support this rather broad claim, especially w.r.t. the following aspects
- the authors promised "uncertainty estimates" in the author response for the results but only provide standard deviation for the dataset. It remains unclear how significant the results are.
- comparison to prior work: The authors discuss prior work w.r.t. Efficient Transformers and Character-level Models, but miss to compare experimentally.
- The authors evaluate as they state "custom" patent classification dataset, but the general claim should entail an evaluation of diverse benchmarks. That the evaluation is limited to this patent dataset is only mentioned in the experimental section.
- additional ablations (e.g. providing the results w.r.t. same number of heads in the paper) and insights would also support the claim.


**Resubmission Of Major Revision:**

The authors may consider submitting a major revision at a later time.